# Catecholaminergic challenge uncovers distinct Pavlovian and instrumental mechanisms of motivated (in)action

Jennifer C Swart[1]*, Monja I Froböse[1], Jennifer L Cook[1,2], Dirk EM Geurts[1,3], Michael J Frank[4,5], Roshan Cools[1,3]†, Hanneke EM den Ouden[1]*†

[1]Donders Institute for Brain, Cognition and Behaviour, Radboud University, Nijmegen, The Netherlands; [2]School of Psychology, University of Birmingham, Birmingham, United Kingdom; [3]Department of Psychiatry, Radboud University Medical Center, Nijmegen, The Netherlands; [4]Department of Cognitive, Linguistic and Psychological Sciences, Brown University, Providence, United States; [5]Brown Institute for Brain Sciences, Brown University, Providence, United States

**Abstract** Catecholamines modulate the impact of motivational cues on action. Such motivational biases have been proposed to reflect cue-based, 'Pavlovian' effects. Here, we assess whether motivational biases may also arise from asymmetrical instrumental *learning* of active and passive responses following reward and punishment outcomes. We present a novel paradigm, allowing us to disentangle the impact of reward and punishment on instrumental learning from Pavlovian response biasing. Computational analyses showed that motivational biases reflect both Pavlovian and instrumental effects: reward and punishment cues promoted generalized (in)action in a Pavlovian manner, whereas outcomes enhanced instrumental (un)learning of chosen actions. These cue- and outcome-based biases were altered independently by the catecholamine enhancer methylphenidate. Methylphenidate's effect varied across individuals with a putative proxy of baseline dopamine synthesis capacity, working memory span. Our study uncovers two distinct mechanisms by which motivation impacts behaviour, and helps refine current models of catecholaminergic modulation of motivated action.

*For correspondence: j.swart@donders.ru.nl (JCS); h.denouden@donders.ru.nl (HEMdO)

†These authors contributed equally to this work

## Introduction

Catecholamine (i.e. dopamine and noradrenaline) transmission has long been implicated in key aspects of adaptive behaviour, including learning, action, and motivation. Deficits in these aspects of adaptive behaviour are observed in a wide range of neuropsychiatric disorders, such as attention deficit hyperactivity disorder, Parkinson's disease, and addiction (*Dagher and Robbins, 2009*; *Prince, 2008*; *Skolnick, 2005*), and many of those deficits can be treated with catecholaminergic drugs (*Faraone and Buitelaar, 2010*; *Wigal et al., 2011*). While overwhelming evidence implicates catecholamines in both motivated activation and motivated learning of behaviour (*Bromberg-Martin et al., 2010*; *Robbins and Everitt, 1996*; *Wise, 2004*), their respective contributions are still highly debated. In this study, we dissect the contribution of catecholamines to motivational biases in behavioural activation and learning.

The neuromodulator dopamine has been linked particularly strongly to behavioural activation in the context of reward (*Taylor and Robbins, 1986*; *1984*), putatively by amplifying the perceived benefits of action over their costs (*Collins and Frank, 2014*; *Niv et al., 2007*). This behavioural activation to reward-predicting cues is likely to be, at least partly, Pavlovian in nature, with the conditioned cues eliciting innately specified responses (*Figure 1A*). The Pavlovian nature of these

**eLife digest** When we see a threat, we tend to hold back. When we see a reward, we have a strong urge to approach. Most of the time, these hardwired tendencies – or biases – are the right thing to do. However, our behaviour is not all hardwired; we can also learn from our previous experiences. But might this learning be biased too? For example, we might be quicker to believe that an action led to a reward, because actions often do bring rewards. Conversely, we might be less likely to attribute a punishment to having held back, because holding back usually helps us to avoid punishments.

Swart et al. have now tested whether rewards and punishments influence our actions solely via hardwired behavioural tendencies, or whether they also bias our learning. That is, are we biased to learn that taking action earns us rewards, while holding back spares us punishments? Previous work has shown that chemical messengers in the brain called catecholamines help us to take action when we anticipate a reward. Swart et al. therefore also examined whether catecholamine levels contribute to any bias in learning.

One hundred young healthy adults twice performed a task in which they could earn rewards and avoid losses by taking or withholding action. By using a mathematical model to work out what influenced the choices made by the volunteers, Swart et al. found that rewards and punishments did indeed bias learning. Moreover, this learning bias became stronger when the volunteers took methylphenidate (also known as Ritalin), a drug that increases catecholamine levels and which is used to treat ADHD and narcolepsy. The volunteers varied markedly in how strongly methylphenidate affected their choices. This emphasises how important it is to account for differences between people when evaluating the effects of medication.

Motivations are what get us going and keep us going. The findings of Swart et al. mean that we now have a better understanding of how motivations, such as desired rewards or unwanted punishments, influence our behaviour. A future challenge is to understand how we can overcome these motivations when they work against us, such as in addiction or obesity.

motivational biases has been demonstrated using Pavlovian-instrumental transfer (PIT) paradigms (*Estes and Skinner, 1941*; *Estes, 1943*). In PIT, conditioned cues elicit innately specified responses that may potentiate (or interfere with) instrumental responding, e.g. appetitive cues promote active responding (appetitive PIT), whereas aversive cues increase behavioural inhibition (aversive PIT; *Davis and Wright, 1979*; *Huys et al., 2011*). Enhanced dopamine increases appetitive PIT (*Wyvell and Berridge, 2000*), while appetitive PIT is lowered when striatal dopamine is reduced (*Dickinson et al., 2000*; *Hebart and Gläscher, 2015*; *Lex and Hauber, 2008*). Striatal dopamine has also been linked to controlling aversively motivated behaviour (*Faure et al., 2008*; *Lloyd and Dayan, 2016*). Together, these results show that appetitive cues promote activation and aversive cues promote inhibition in a Pavlovian manner, mediated by the dopamine system.

While Pavlovian response biases can be helpful in reducing computational load by shaping our actions in a hardwired manner, they are inherently limited because of their general nature (*Dayan et al., 2006*). In order to select the best action in a specific environment, instrumental systems allow organisms to adaptively learn action-outcome contingencies, by assigning value to actions that in the past have led to good outcomes, while reducing value of actions that led to negative outcomes (*Dickinson and Balleine, 1994*; *Rescorla and Wagner, 1972*; *Robbins and Everitt, 2007*). Pavlovian and instrumental learning are often presented as a dichotomy, whereby cue-based, Pavlovian effects are solely responsible for motivational biases, while adaptive 'rational' choice results from instrumental learning. For example, multiple recent studies showing that reward or punishment cues bias action, eliciting appetitive activation and/or aversive inhibition, have been interpreted specifically in terms of a Pavlovian response bias (for a review see *Guitart-Masip et al., 2014a*).

We hypothesised that these motivational biases of behavioural activation may also arise from asymmetrical, or biased, instrumental learning (*Figure 1B*), in addition to Pavlovian response biases. Such biases in learning, like response biases, may reflect predominant statistics of the environment.

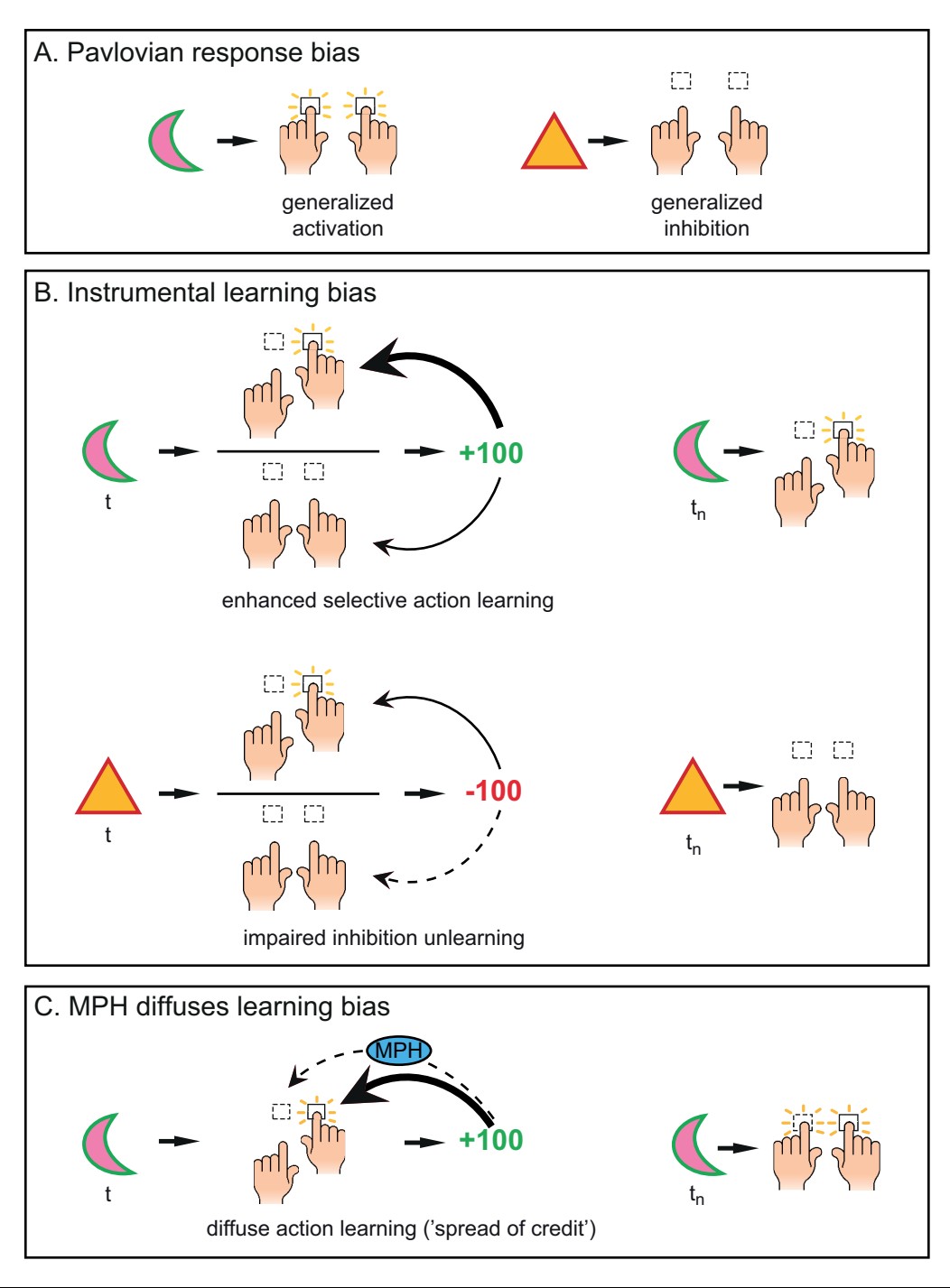

**Figure 1.** Distinct mechanisms by which motivational valence may bias behavioural activation. (**A**) Pavlovian response bias: appetitive cues (green edge) elicit generalized behavioural activation ('Go'), whereas aversive cues (red edge) elicit behavioural inhibition ('NoGo'). This Pavlovian response bias is introduced in model M3a as the parameter π (c.f. *Figure 3*). (**B**) Instrumental learning bias: rewarding outcomes (upper panel) facilitate learning of action ('Go', thick arrow) relative to inaction ('NoGo', thin arrow). Thus, learning effects at the individual trials *t* will result in a cumulative selective increase of the rewarded action on later trials $t_n$. Punishment outcomes (lower panel) hamper the unlearning of inaction ('NoGo', dashed arrow) relative to action ('Go', solid arrow), resulting in sustained inaction. Neutral outcomes are equally well associated with actions and inactions, and are not illustrated here. The instrumental learning bias is introduced as the parameter κ in model M3b (c.f. *Figure 3*). We assess whether these two mechanisms (i) act in parallel, and (ii) are modulated by the catecholamine system. To test the

*Figure 1 continued on next page*

*Figure 1 continued*

latter, we administered methylphenidate (MPH), which prolongs the effects of catecholamine release via blockade of the catecholamine receptors. We first assess whether MPH affects the strength of the Pavlovian response bias, introduced as the parameter $\pi_{MPH}$ in model M5a, and instrumental learning bias, implemented as the parameter $\kappa_{MPH-selective}$ in model M5b (c.f. *Figure 5*). (C) We hypothesise that prolonged effects of dopamine release following reward outcomes will reduce (temporal) specificity, leading to spread of credit: Credit is assigned to other recent actions (thin arrow), in addition to the performed (and rewarded) Go response (thick arrow), resulting in additional learning of the alternative (not-performed) Go response. This MPH-induced diffuse learning bias is implemented by the parameter $\kappa_{MPH-diffuse}$ in model M5c (c.f. *Figure 5*).

For example, we might be quicker to believe that an action led to a reward, because actions often cause rewards. However, we may not attribute a punishment to having held back, because holding back usually helps avoid a punishment. Such an instrumental learning bias may arise from a circuitry where reinforcers are more potent at driving learning following active 'Go' than inactive 'NoGo' actions. This means that Go responses (relative to NoGo responses) are easier to learn and unlearn following reward and punishment respectively. This instrumental learning bias would predict that Go responses that elicited a reward are more likely to be repeated (i.e. better learned) than NoGo responses that elicited a reward. Similarly, Go responses that elicited a punishment are relatively less likely to be repeated (i.e. better unlearned) than NoGo responses that elicited a punishment. These instrumental learning and Pavlovian response biasing accounts of motivated (in)action could not be dissociated in earlier studies (*Cavanagh et al., 2013*; *Guitart-Masip et al., 2014b*; *2012*), because they allowed for only a single Go response: With only one response option, general activation of action cannot be disentangled from facilitated learning of a specific response. In our proposed framework, motivational biases in behavioural (in)activation are likely the result of a combination of Pavlovian response biasing plus an asymmetry in instrumental learning of Go and NoGo responses (*Figure 1*).

At the neurobiological level, this hypothesis arises from current theorizing about the mechanism of action of reinforcement-related changes in dopamine. Specifically, a potential substrate for this proposed learning asymmetry could be provided by the striatal dopamine system, which is notably involved in instrumental learning via modulation of synaptic plasticity (*Collins and Frank, 2014* for review and models). In this framework, dopamine bursts elicited by better than expected outcomes reinforce the actions that led to these outcomes (*Montague et al., 2004*; *Schultz et al., 1998*; *Schultz et al., 1997*) via long-term potentiation (LTP) in the striatal direct 'Go' pathway (*Frank et al., 2004*). The temporal specificity of the phasic dopamine bursts allows for assigning credit to the most recent action, by potentiating the recently active striatal neurons. Due to the LTP in the 'Go' pathway, rewards may be more effective in reinforcing neurons coding for active Go responses than NoGo responses. Conversely, dopamine dips elicited by worse-than-expected outcomes (*Matsumoto and Hikosaka, 2009*; *Tobler et al., 2005*) lead to long-term depression (LTD) of the 'Go' pathway and LTP in the 'NoGo' pathway, making it less likely that the same cue would elicit an active than inactive response next time. In short, the striatal system is biased to attribute rewards and punishments to active Go responses, which ecologically may be more commonly the cause of observed outcomes. The implication of this is that is easier to learn to take action based on reward, but easier to withhold making an action based on punishment.

A key additional prediction of this model is that prolonging the presence of dopamine, e.g. by blocking dopamine reuptake with methylphenidate, would lead to a spread of credit assignment (*Figure 1C*). Here, credit is assigned to striatal neurons that were recently active, due to recent actions that did not actually lead to the current reward and phasic dopamine burst ('spread of effect'; *Thorndike, 1933*). In this framework, the dopamine system can produce biased motivated behaviour due to (i) direct Pavlovian biases (e.g. predicted rewards potentiate the Go pathway during action selection), and (ii) disproportionate potentiation of instrumental learning of Go actions that (recently) led to reward. Put more simply, (i) dopamine bursts prompted by reward-predicting cues can potentiate activation of the Go pathway, giving rise to the cue-based, Pavlovian activation, and (ii) dopamine bursts prompted by reward outcomes can potentiate plasticity within the Go pathway, making rewards more effective in reinforcing Go responses than NoGo responses.

In this study, we aimed to assess whether biases in instrumental learning contribute to the pharmaco-computational mechanisms subserving well-established reward/punishment biases of motivated (in)action. To dissociate biased instrumental learning from Pavlovian response biases, we developed a novel experimental paradigm including multiple active response options (*Figure 2*), and combined this task with a catecholamine challenge (catecholamine reuptake blocker methylphenidate - MPH). We tested the following hypotheses: (i) cue-valence (appetitive vs. aversive cues) biases action in a Pavlovian manner, whereas outcome-valence (reward vs. punishment) biases instrumental learning of Go vs. NoGo actions; (ii) blocking the catecholamine reuptake with MPH enhances the strength of the Pavlovian response bias as a result of prolonged dopamine release to reward cues; (iii) MPH reduces the specificity of credit assignment to specific actions that elicited rewards, as the prolonged DA release to reward outcomes would spread credit to non-chosen active actions (*Figure 1*).

Finally, MPH prolongs the effects of catecholamine release by blocking the reuptake of catecholamines, without stimulating release or acting as a receptor (ant)agonist (e.g. *Volkow et al., 2002*). Accordingly, it is likely that the effect of MPH on catecholamine-dependent function is a function of dopamine synthesis capacity and release. Simply put, if there is no release, there is no reuptake to block. To assess these potential sources of individual variability in MPH effects, we took into account two measures that have been demonstrated with PET to relate to dopamine baseline function: working memory span for its relation to striatal dopamine synthesis capacity (*Cools et al., 2008*; *Landau et al., 2009*) and trait impulsivity for its relation to dopamine (auto)receptor availability (*Buckholtz et al., 2010*; *Kim et al., 2014*; *Lee et al., 2009*; *Reeves et al., 2012*), and collected a large sample (N = 106) to expose individual differences.

## Results

Healthy participants performed a motivational Go/NoGo learning task, in which cue valence (Win vs. Avoid cue) is orthogonalized to the instrumental response (Go vs. NoGo). During this task, subjects need to learn for each of 8 cues to make a Go or NoGo response, and by making the correct response subjects are rewarded for Win cues (green edge) and avoid punishment for the Avoid cues (red edge) in 80% of the time. Crucially, in contrast to task designs in previous studies (*Guitart-Masip et al., 2014a*), in this novel paradigm subjects could make either of two Go responses (press left vs. right) or withhold responding (NoGo; *Figure 2A–D*). Including two Go response options enabled us to tease apart general activation/inhibition related to the Pavlovian response bias and specific action learning related to the instrumental learning bias using computational models and behavioural analyses.

### Motivational Valence affects (in)correct action

Subjects successfully learned this difficult task, in which they needed to identify the correct response out of 3 options (Go-left/Go-right/NoGo) for eight different cues, as evidenced by increased Go responding to cues indicating the need to Go vs. NoGo (Required Action: $X^2(1)=624.3$; p<0.001; *Figure 2E,F*). In other words, subjects were able to adjust Go responding to the required action. As expected, cue valence also influenced Go responding (Valence: $X^2(1)=40.0$; p<0.001), reflecting a motivational bias in responding. Overall subjects made more Go responses for Win than Avoid cues. The effect of cue valence was highly significant for both Go and NoGo cues (Go cues: $X^2(1)=37.5$, p<0.001; NoGo cues: $X^2(1)=13.3$, p<0.001), though marginally stronger for the Go cues (Required Action x Valence: $X^2(1)=3.1$; p=0.076). Because each Go cue was associated with only one correct Go response, we confirmed that this motivational bias was present for both correct and incorrect Go responses. Subjects made more Go responses to Win than avoid cues for both correct (Valence: $X^2(1)=26.1$, p<0.001) and incorrect (Valence: $X^2(1)=25.6$, p<0.001) Go responses. Next, we tested the hypothesis that this motivational bias arose from a combination of a Pavlovian response bias and biased instrumental learning (*Figure 1A–B*).

### Computational modelling: disentangling Pavlovian response bias and instrumental learning bias

We used a computational modelling approach to quantify latent processes that we hypothesised to underlie the behavioural performance. Specifically, our first aim was to disentangle the contribution

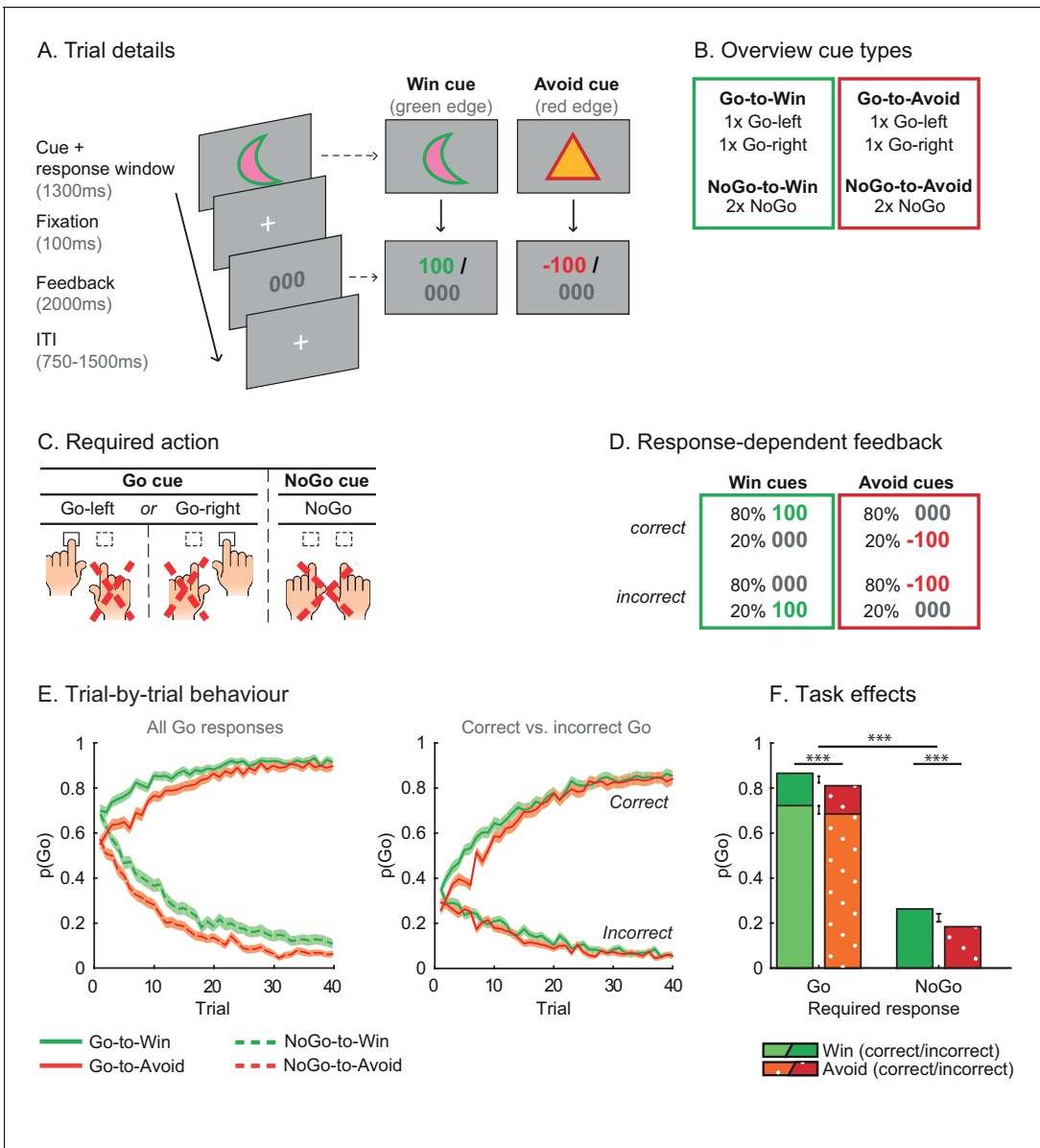

**Figure 2.** Motivational Go/NoGo learning task and performance. (**A**) On each trial, a Win or Avoid cue appears on screen. Subjects can respond during cue presentation. Response-dependent feedback follows. (**B**) In total eight cues are presented for which the correct response needs to be learned. (**C**) Each cue has only one correct response (Go-left, Go-right, or NoGo), which subjects can learn from the feedback. (**D**) Feedback is probabilistic. Correct responses are followed by reward (Win cues) or a neutral outcome (Avoid cues) in 80% of the time and by a neutral outcome (Win cues) or punishment (Avoid cues) otherwise. For incorrect responses, these probabilities are reversed. (**E**) Trial-by-trial proportion of Go responses (±SEM) for Go cues (solid lines) and NoGo cues (dashed lines), collapsed over Placebo and MPH. Left: All cue types. From the first trial onwards, subjects made more Go responses to Win vs. Avoid cues (i.e. green lines are above red lines), reflecting the motivational bias. Additionally, subjects clearly learn whether to make a Go response or not (proportion of Go responses increases for Go cues and decreases for NoGo cues). Right: Go cues only. For the Go cues, a Go response could be either correct or incorrect. The motivational bias is present in both correct and incorrect Go responses, but incorrect Go responses are unlearnt. Note that the total p(Go) in this plot sums up to the solid lines in the left plot. (**F**) Mean (±SED) proportion Go responses. Proportion Go responses is higher for Go vs. NoGo cues, indicative of task learning. Additionally, subjects made more correct and incorrect Go responses to Win vs. Avoid cues. Source data of task performance are available in *Figure 2—source data 1*.

The following source data and figure supplement are available for figure 2:

**Source data 1.** Source data for task performance under MPH and placebo.

**Figure supplement 1.** Individual traces (black lines) and group average (coloured lines) of correct and incorrect Go responses using a sliding average of 5 trials.

of Pavlovian response biases and instrumental learning biases to the observed valence effect in behaviour. To this end we extended a simple reinforcement learning model using hierarchical Bayesian parameter estimation. We developed five nested base models (M1, M2, M3a, M3b, M4) with increasing complexity to assess whether additional parameters explained the observed data better, while penalizing for increasing complexity.

In all models, the probability of each response is estimated based on computed action weights. In the simplest model (M1) the action weights are fully determined by the learned action values ($Q$-values). Action values are updated with the prediction error, i.e. the deviation of the observed outcome from the expected outcome (standard 'delta-rule' learning; *Rescorla and Wagner, 1972*). M1 contains two free parameters: a learning rate ($\epsilon$) scaling impact of the prediction-error, and feedback sensitivity ($\rho$) scaling the outcome value. Next, to allow for a non-selective bias in Go responses unrelated to valence, a go bias parameter ($b$) is added to the action weights of Go responses in M2. This parameter simply captures how likely people are to make a 'Go' response overall.

In this task, we explicitly instructed the cue valence, by colouring the edge of each cue, where green signalled that subjects could win a reward, while red signalled they had to avoid a punishment (*Figure 2A*). As a consequence, we observed an effect of the instructed cue valence on Go responses already from the first trial onwards (*Figure 2E*), implying a motivational bias before learning could occur, which is therefore likely Pavlovian in nature. To assess this Pavlovian response bias, cue values are added to the action weights in M3a. In this model positive (negative) Pavlovian values increase (decrease) the action weight of Go responses, where $\pi$ scales the weight of the Pavlovian values (*Cavanagh et al., 2013*; *Guitart-Masip et al., 2014b*; *2012*). Thus, the Pavlovian bias parameter increases the probability of all Go responses for Win cues and decreases the probability of all Go responses for Avoid cues.

In M3b we assessed whether a motivational learning bias affects behaviour. Specifically, we included an instrumental learning bias parameter ($\kappa$), to assess whether reward is more effective in reinforcing Go responses than NoGo responses, whereas punishment is less effective in unlearning NoGo responses than Go responses. This biased learning parameter indexes the degree to which the *specific* Go response that elicited a reward would be relatively more likely to be repeated in subsequent trials, resulting in increased instrumental learning of Go responses for reward. Note that earlier studies used only a single Go response and could thus not dissociate this specific learning vs. Pavlovian bias account. In addition to this effect on learning from rewards, $\kappa$ indexes the degree to which punishment is biased to potentiate activity in the NoGo versus Go pathway, thus biasing unlearning to be more effective after Go responses than after NoGo responses, (i.e., making punishment-based avoidance learning of NoGo responses more difficult than punishment-based avoidance learning of Go responses; *Figure 1B*). Because the Pavlovian and instrumental learning bias might explain similar variance in the data, we tested model M4, where we included both $\pi$ and $\kappa$ to test whether there was evidence for the independent presence of both the instrumental learning bias and the Pavlovian response bias.

Stepwise addition of the go bias (Appendix 5), Pavlovian response bias and instrumental learning bias parameter improved model fit, as quantified by Watanabe-Akaike Information Criteria (WAIC; *Figure 3*; *Table 1*). The Pavlovian bias parameter estimates ($\pi$) of the winning model M4 were positive across the group (96.4% of posterior distribution >0). The Pavlovian bias estimates were modest across the group (*Figure 3*; *Table 1*), and showed strong individual variability (*Figure 3—figure supplement 2*; *Figure 3—figure supplement 3*). This strong inter-individual variability is consistent with previous reports, e.g. *Cavanagh et al. (2013)*, who show that differences in the strength of the Pavlovian bias is inversely predicted by EEG mid-frontal theta activity during incongruent relative to congruent cues, putatively reflecting the ability to suppress this bias on incongruent trials. The further improvement of model fit due to the instrumental learning bias parameter (M3a vs. M4) provides clear evidence for the contribution of biased action learning on top of the Pavlovian response bias described in previous studies. The biased instrumental learning parameter estimates were also positive across the group (100% of posterior distribution >0). In other words, in the winning model, the motivational bias, as reflected by an increase in Go responses to Win relative to Avoid cues, is explained by the presence of both a Pavlovian response bias and biased instrumental learning. *Figure 3* and accompanying Figure supplements illustrate the model predictions and parameter estimates.

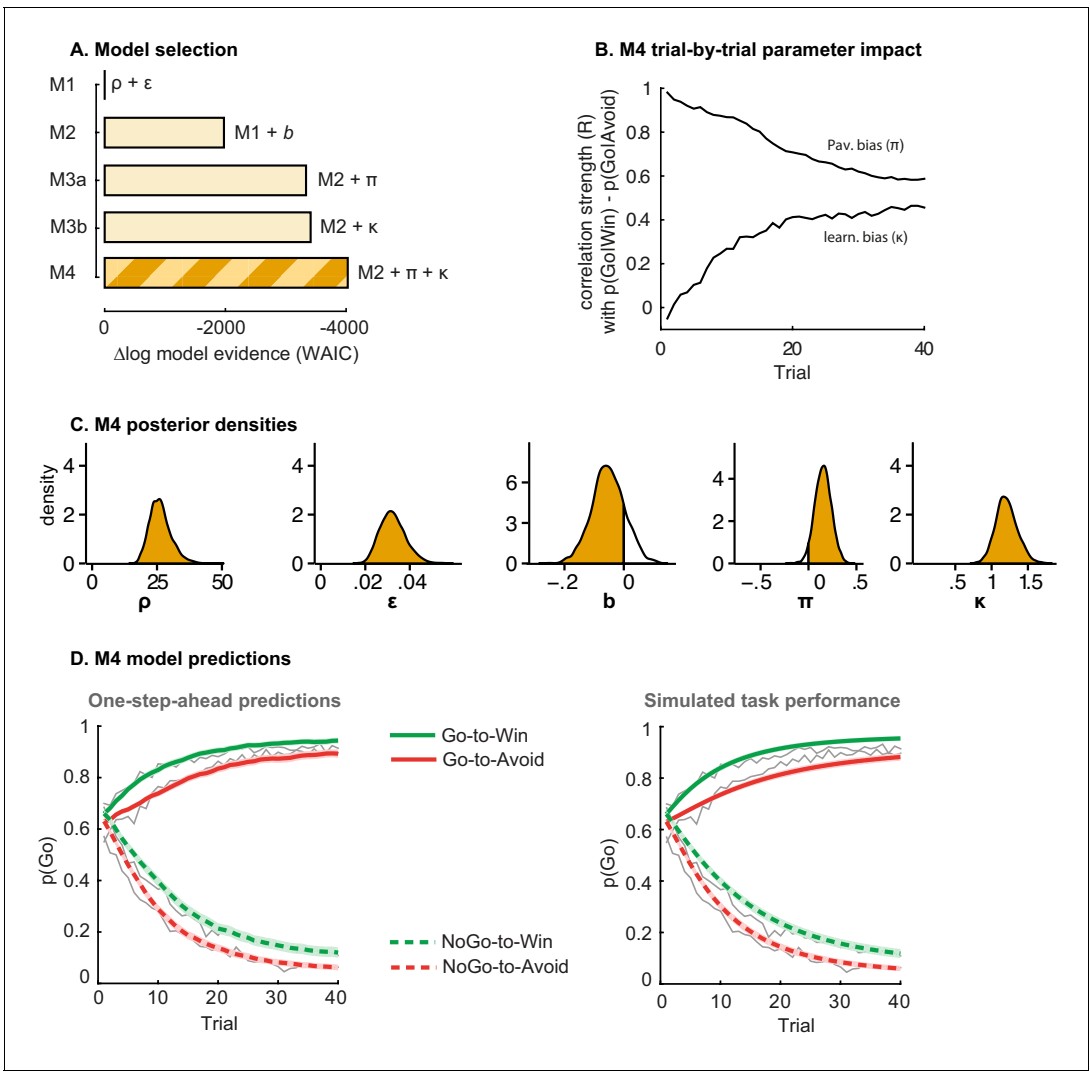

**Figure 3.** Model evidence and parameter inference of base models. (A) Model evidence, relative to simplest model M1, clearly favours M4. The simplest model M1 contains a feedback sensitivity (ρ) and learning rate (ε) parameter. Stepwise addition of the go bias (b), Pavlovian bias (π; *Figure 1A*), and instrumental learning bias (κ; *Figure 1B*) parameter improves model fit, quantified by WAIC (estimated log model evidence). Lower (i.e. more negative) WAIC indicates better model fit. (B) Temporal dynamics of the correlation between the motivational bias parameters (M4) and the predicted motivational bias, i.e. probability to make a Go response to Win relative to Avoid cues. The impact of the Pavlovian bias (π) on choice decreases over time (although, importantly, the parameter itself remains constant). This is because the instrumental values of the actions are learnt and thus will increasingly diverge. As a result, π is less and less 'able' to tip the balance in favour of the responses in direction of the motivational bias (i.e. it can no longer overcome the difference in instrumental action values). In contrast, the impact of κ on choice increases over time, reflecting the cumulative impact of biased learning (also *Figure 3—figure supplement 2*). (C) Posterior densities of the winning base model M4. Appendix 5 shows posterior densities for all models. (D) One-step-ahead predictions and posterior predictive model simulations of winning base model M4 (coloured lines), to assess whether the winning model captures the behavioural data (grey lines). Both absolute model fit methods use the fitted parameters to compute the choice probabilities according to the model. The one-step-ahead predictions compute probabilities based on the history of each subject's actual choices and outcomes, whereas the simulation method generates new choices and outcomes based on the response probabilities (see Materials and methods for details). Both methods capture the key features of the data, i.e. responses are learnt (more 'Go' responding for 'Go' cues relative to 'NoGo' cues) and a motivational bias (more Go responding for Win relative to Avoid cues). We note that the model somewhat underestimates the initial Pavlovian bias (i.e. difference in Go responding between Win and Avoid trials is, particularly trial 1–2), while it overestimates the Pavlovian bias on later trials. This is likely the result from the fact that while the modelled Pavlovian bias parameter (π) is constant over time, the impact of the Pavlovian stimulus values weakens over time, as the subjects' confidence in the instrumental action values increases. Interestingly, notwithstanding the constancy of the Pavlovian bias parameter, we do capture some of these dynamics as *Figure 3B* shows that the impact of the Pavlovian bias on choice decreases over time. Source data of M4 simulated task performance are available in *Figure 3—source data 1*.

The following source data and figure supplements are available for figure 3:

*Figure 3 continued on next page*

*Figure 3 continued*

**Source data 1.** Source data for model M4 simulated task performance.
**Figure supplement 1.** Subject traces of model M4 (green/red) overlaid on observed behavior (black).
**Figure supplement 2.** Illustration of the behavioural effects associated with the Pavlovian bias and instrumental learning bias parameters.
**Figure supplement 3.** M4 subject-level parameters in model space (i.e. untransformed).

## MPH enhances effect of cue valence proportional to working memory span

Next, we asked whether acute administration of MPH altered the motivational bias. As noted above, the effects of dopaminergic drugs often depend on baseline dopamine function. We therefore used two neuropsychological measures that have been shown to predict baseline dopamine function using PET: working memory span, predictive of baseline dopamine synthesis capacity (*Cools et al., 2008*; *Landau et al., 2009*), and trait impulsivity, predictive of D2 autoreceptor availability (*Buckholtz et al., 2010*; *Kim et al., 2014*; *Lee et al., 2009*; *Reeves et al., 2012*). Importantly, both working memory span and trait impulsivity predict dopaminergic drugs effects on various cognitive functions (*Clatworthy et al., 2009*; *Cools et al., 2009*; *2007*; *Frank and O'Reilly, 2006*; *Gibbs and D'Esposito, 2005*; *Kimberg et al., 1997*; *Zimet et al., 1988*).

MPH enhanced the effect of cue valence on Go responding proportional to working memory span (Valence x Drug x Listening Span: $X^2(1)=5.9$; p=0.016; *Figure 4B*), in the absence of a Valence x Drug effect across the group (Valence x Drug: $X^2(1)=1.5$; p=0.221; *Figure 4A*). While high-span subjects showed a drug-induced increase in motivational bias (MPH versus placebo increased Go responding to Win vs. Avoid cues), low-span subjects showed a drug-induced decrease in motivational bias. This span-dependent bias emerged under MPH ($X^2(1)=4.6$, p=0.032), and was not significant under placebo ($X^2(1)=0.9$, p=0.335; *Figure 4—figure supplement 1*).

A break-down of this effect into correct and incorrect responses revealed that it was driven by *incorrect* Go responses (Valence x Drug x Listening Span: $X^2(1)=11.9$, p<0.001; *Figure 4C*). MPH did not significantly affect the *correct* Go responses (Valence x Drug x Listening Span: $X^2(1)=2.0$, p=0.152). In other words, higher span subjects were more likely to make Go responses to Win cues under MPH, but this Go response was more likely to be incorrect. We reasoned that an enhanced learning bias would manifest primarily in increased *correct* Go responses to Win cues (i.e. the correct responses are better learned), while an enhanced Pavlovian bias or diffusion of credit assignment would manifest in increased *correct* and *incorrect* Go responses to Win cues (due to overall action invigoration and potentiation respectively). Thus, we expected that the altered valence effect on

**Table 1.** Base models. Median [25–75 percentile] of subject-level parameter estimates in model space. See Appendix 5 for subject-level / top-level parameters in sampling space (i.e. untransformed). Absolute WAIC is reported at the top as the estimate of model evidence, where a smaller WAIC indicates higher evidence.

| | Base models | | | | |
|---|---|---|---|---|---|
| | **M1** | **M2** | **M3a** | **M3b** | **M4** |
| WAIC | 71014 | 69038 | 67678 | 67602 | 66987 |
| $\rho$ | 42.7 [19.3 79.8] | 41.6 [18.7 72.4] | 35.2 [15.8 66.4] | 33.4 [13.9 59.8] | 32.5 [14.9 56.4] |
| $\epsilon_0$ | 0.013 [0.008 0.059] | 0.015 [0.008 0.054] | 0.017 [0.009 0.064] | 0.022 [0.010 0.070] | 0.021 [0.010 0.063] |
| $b$ | | −0.25 [−0.45 0.04] | −0.25 [−0.46 0.04] | .01 [−0.33 0.27] | −0.03 [−0.29 0.19] |
| $\pi$ | | | 0.47 [0.02 1.00] | | 0.12 [−0.29 0.70] |
| $\epsilon_{\text{ rewarded Go}}(\epsilon_0+\kappa)$ | | | | 0.037 [0.016 0.122] | 0.034 [0.016 0.109] |
| $\epsilon_{\text{ punished NoGo}}(\epsilon_0-\kappa)$ | | | | 0.006 [0.002 0.014] | 0.008 [0.003 0.022] |

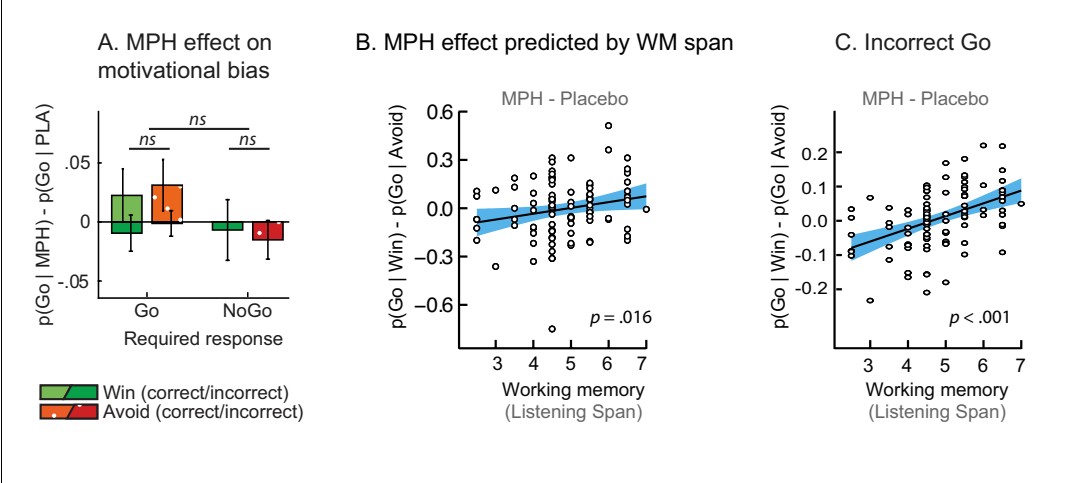

**Figure 4.** MPH-induced changes in motivational bias (i.e. proportion of Go responses to Win relative to Avoid cues). (**A**) Mean (±SED) proportion Go responses under MPH relative to Placebo. MPH did not significantly alter the motivational bias across the group (p=0.22; *ns* indicates p>0.05). (**B**) MPH increased the motivational bias in high span subjects, yet decreased it in low span subjects (R = 0.21; p=0.016). (**C**) MPH altered the motivational bias particularly for *incorrect* Go proportional to working memory span (incorrect Go: p<0.001; *correct* Go: p=0.152).
The following figure supplement is available for figure 4:

**Figure supplement 1.** Simple effects of MPH-induced changes in motivational bias.

*incorrect* Go responses under MPH can best be attributed to MPH alteration of Pavlovian response bias or diffusion of credit assignment, which we formally test using computational modelling (see below).

In contrast to listening span, trait impulsivity did not significantly predict the effect of MPH on the motivational bias (all p>0.05; see Appendix 3 for an overview of the mixed model effects). We confirmed that the MPH effects were not explained by session effects, i.e. whether MPH was received on the first or second testing day ($X^2$(2)=2.1, p=0.349), nor did the factor Testing day improve model fit ($X^2$(1)=2.0, p=0.162). Finally, we confirmed that including nuisance variables Gender and NLV scores (measuring verbal intelligence), did not improve model fit either ($X^2$(2)=0.4, p=0.815).

## Computational modelling: dissociable effects of MPH on pavlovian response bias and biased instrumental learning

Continuing our modelling approach, we next assessed whether the MPH-induced motivational bias could be attributed to an altered Pavlovian response bias and/or instrumental learning bias. To this end we extended the winning base model M4 into competing models. In M5a we included an MPH-induced Pavlovian bias parameter ($\pi_{MPH}$), to assess whether MPH altered the Pavlovian response bias. Here $\pi_{MPH}$ alters the individual's Pavlovian bias parameter under MPH. In M5b we included an MPH-induced instrumental learning bias ($\kappa_{MPH-selective}$). Thus, M5b tests whether MPH affects the strength of the instrumental learning bias in individuals. We further tested whether MPH might make the learning bias more diffuse, because of its mechanisms of action. Because MPH blocks reuptake, it prolongs dopamine release, such that reinforcement and synaptic potentiation might not be attributed only to the temporally coincident neurons that code for the recently selected action, but could be spread to other actions (*diffuse learning*). To test this hypothesis, M5c contains a MPH-induced diffuse learning bias ($\kappa_{MPH-diffuse}$), where $\kappa_{MPH-diffuse}$ is a learning rate that alters the value of all Go responses following a reward, under MPH (*Figure 1C*) by scaling the prediction error following all rewarded Go responses.

Model fit improved markedly when extending the winning base model M4 with the MPH-induced Pavlovian bias parameter $\pi_{MPH}$ (M5a; *Figure 5*; *Table 2*). Extending M4 with the MPH-induced selective learning bias parameter $\kappa_{MPH-selective}$ (M5b) only slightly increased model fit. Conversely, the MPH-induced diffuse learning bias parameter $\kappa_{MPH-diffuse}$ (M5c) also strongly improved model fit

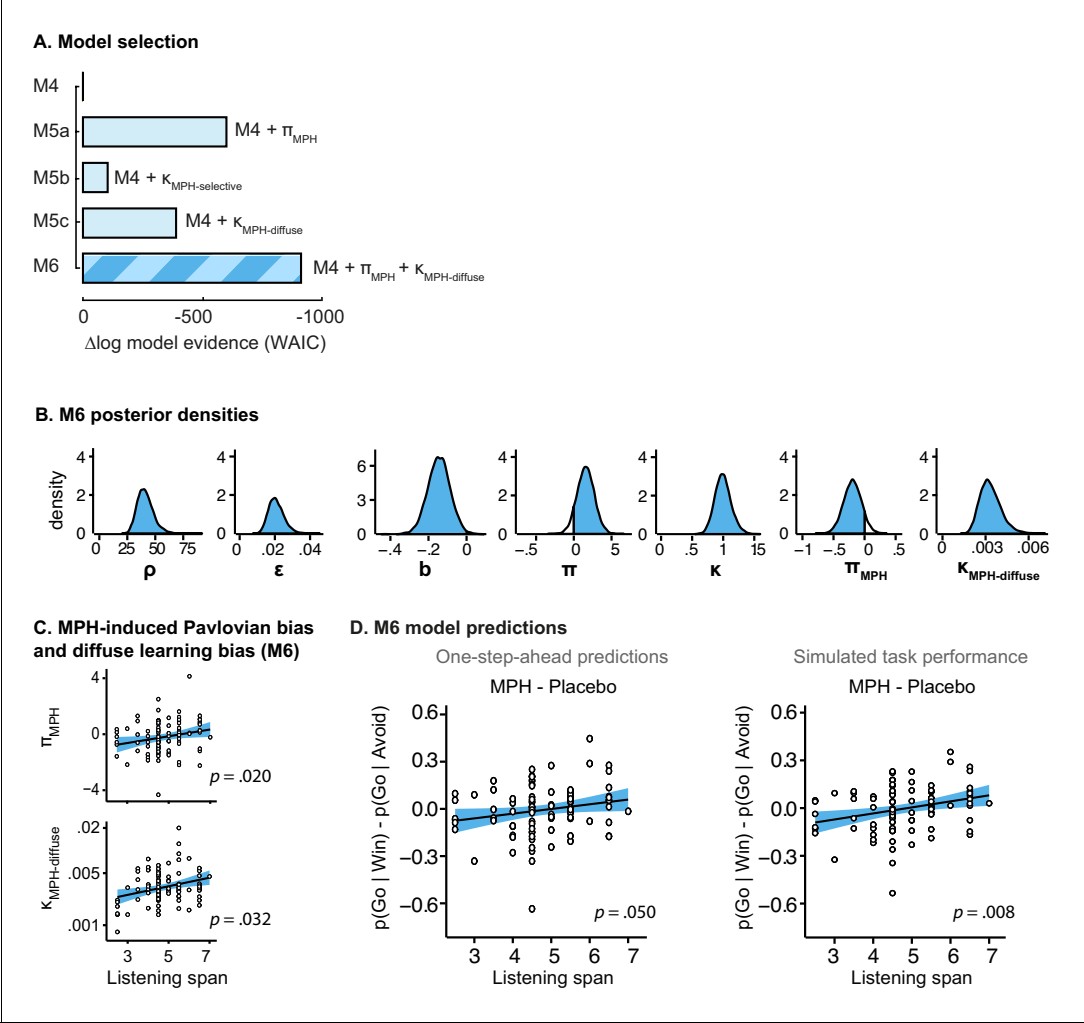

**Figure 5.** Model evidence and parameter inference of extended MPH models. (**A**) Model evidence (WAIC) relative to winning base model M4. We tested whether MPH alters the strength of the Pavlovian response bias ($\pi_{MPH}$; M5a), the instrumental learning bias ($\kappa_{MPH-selective}$; M5b), or has a diffuse effect on the learning bias ($\kappa_{MPH-diffuse}$; M5c; *Figure 1C*). Model selection favoured the composite model M6, including the $\pi_{MPH}$ and $\kappa_{MPH-diffuse}$ parameters. (**B**) Posterior densities of the top-level parameters of M6. (**C**) Subject-level estimates of MPH-induced Pavlovian bias parameter (upper) and the MPH-induced diffuse learning bias parameter (lower; logistic scale) correlated significantly with Listening Span. (**D**) One-step-ahead model predictions and posterior predictive model simulations of M6 using subject-level parameter estimates. The model predictions and simulations echo the observed data, i.e. that the motivational bias correlates positively with working memory span (*Figure 4B*), confirming the winning model M6 captures the MPH-induced increase in Go responses to Win vs. Avoid cues.

The following figure supplements are available for figure 5:

**Figure supplement 1.** Illustration of the behavioural effects of MPH related to the Pavlovian bias and diffuse learning bias parameters.

**Figure supplement 2.** M6 subject-level parameters in model space (i.e. untransformed).

relative to base model M4. This observation is in line with our earlier prediction that the MPH effects are predominantly driven by changes in the proportion of *incorrect* Go responses. Confirming the model comparison results, the MPH modulation of Pavlovian bias and diffuse learning parameters both covaried with Listening Span ($\pi_{MPH}$: $R = 0.25$, p=0.013; $\kappa_{MPH-diffuse}$: $R = 0.28$, p=0.006), while the MPH selective learning bias did not ($\kappa_{MPH-selective}$: $R = -0.01$, p=0.9). In other words, $\kappa_{MPH-selective}$ did not explain our effect of interest and improved model fit relatively weakly.

**Table 2.** MPH models. Median [25–75 percentile] of subject-level parameter estimates in model space. Absolute WAIC is reported as the estimate of model evidence, where a smaller WAIC indicates higher evidence. Biased instrumental learning rate for rewarded Go and punished NoGo responses as computed by $\varepsilon_0 \pm \kappa$ under placebo and by $\varepsilon_0 \pm (\kappa + \kappa_{MPH})$ under MPH. (MPH) indicates the value of that parameter under MPH.

| | Extended MPH models | | | |
| --- | --- | --- | --- | --- |
| | **M5a** | **M5b** | **M5c** | **M6** |
| WAIC | 66383 | 66883 | 66595 | 66069 |
| $\rho$ | 31.2 [14.7 53.6] | 31.6 [15.6 57.0] | 55.8 [19.6 104.8] | 51.9 [20.6 98.7] |
| $\varepsilon_0$ | 0.022 [0.010 0.067] | 0.021 [0.011 0.061] | 0.011 [0.006 0.051] | 0.012 [0.006 0.055] |
| $b$ | −0.04 [−0.33 0.18] | −0.05 [−0.34] | −0.10 [−0.37 0.13] | −0.14 [−0.42 0.10] |
| $\pi$ <br> $\pi$ (MPH) | 0.27 [−0.50. 71] <br> 0.20 [−0.38. 71] | 0.15 [−0.28. 70] | 0.05 [−0.46. 61] | 0.27 [−0.47. 74] <br> −0.05 [−0.70. 50] |
| $\varepsilon_{\text{rewarded Go}}$ <br> $\varepsilon_{\text{rewarded Go}}$ (MPH) | 0.037 [.017. 116] | 0.030 [.018. 103] <br> 0.031 [.016. 104] | 0.018 [.009. 082] | 0.019 [.009. 085] |
| $\varepsilon_{\text{punished NoGo}}$ <br> $\varepsilon_{\text{punished NoGo}}$ (MPH) | 0.009 [.004. 030] | 0.009 [.003. 021] <br> 0.008 [.002. 021] | 0.004 [.002. 013] | 0.005 [.002. 017] |
| $\varepsilon_{\text{diffuse}}$ (MPH) | | | 0.002 [.002. 004] | 0.003 [.002. 004] |

To assess whether $\pi_{MPH}$ and $\kappa_{MPH\text{-diffuse}}$ explained unique Listening Span-dependent effects of MPH (i.e. whether there was evidence for both of these effects), we constructed a composite model (M6) containing both effects. Model comparison showed that indeed this composite model explained the data best (*Figure 5*). In this model, both parameters again significantly varied proportional to Listening Span ($\pi_{MPH}$: $R = 0.24$, p=0.020; $\kappa_{MPH\text{-diffuse}}$: $R = 0.22$, p=0.032; *Figure 5*).

Taken together, these modelling results attribute the MPH-induced motivational bias partly to an altered Pavlovian response bias ($\pi_{MPH}$), and partly to a reward-driven diffusion of credit during instrumental learning ($\kappa_{MPH\text{-diffuse}}$). In other words, MPH (i) alters the impact of *cue* valence on action, which is present and persists from the first trial onward, and (ii) alters the impact of rewarding *outcomes* on the learning of actions, which fully depends on and evolves with experience. Following a reward, the effect of $\kappa_{MPH\text{-diffuse}}$ is to increase the value of incorrect Go responses in addition to the correct Go response.

Finally, we tested whether our best fitting model was sufficient to reproduce the key features of the data. This is important because model selection only provides relative, but not absolute evidence for the winning model (e.g., *Nassar and Frank, 2016*). We used two approaches to compute the post hoc absolute model fit, namely data simulation and 'one-step-ahead' model predictions. In the simulation method, the first choice is simulated based on the initial values; the corresponding outcome used for learning; the next choice is simulated based on the updated, learned values; and so on. Thus, this simulation method ignores any subject-specific sequential/history effects to determine the current choice probability. Therefore, this can result in choice/outcome sequences that diverge completely from the subjects' actual experiences. Violating the subject-specific choice and outcome history will change the learning effects, making this method less robust in generating the exact learning effects compared to experience-*in*dependent effects. We therefore included a second absolute model fit method that does take into account the subjects' choice and outcome histories: the post-hoc absolute fit method (also known as 'one-step-ahead prediction'; *Pedersen et al., 2016*; *Steingroever and Wagenmakers, 2014*). Here, the initial choice probabilities are determined based on the initial values. For each subsequent trial, the choice probabilities are determined based on the learned values using the actual (subject's) choices and outcomes on all preceding trials. We used both methods as the strongest test providing converging evidence that the models could capture the observed results.

Using both absolute model fit methods, we simulated choices for each individual, using model M6 with each individual's parameter estimates. Both methods confirmed that M6 can capture the observed effects, replicating the Listening Span dependent effect of MPH on choice, where MPH increased Go responses to Win vs. Avoid cues more in higher span subjects (simulations: $R = 0.27$,

p=0.008; one-step-ahead: $R = 0.20$, p=0.050; *Figure 5*). These simulations echo the results reported above, demonstrating the MPH-induced Pavlovian bias parameter $\pi_{MPH}$ and diffuse learning bias $\kappa_{MPH\text{-}diffuse}$ are sufficient to both explain and predict the span-dependent MPH-induced increase in Go responses to Win vs. Avoid cues. *Figure 5* and accompanying Figure supplements illustrate the model predictions and parameter estimates.

## Discussion

Motivational biases of behaviour are well established: Reward biases towards action, punishment towards inaction. In this study, we had two goals. First, we aimed to assess whether these motivational biases arise from biases in instrumental learning in addition to Pavlovian response biases. Second, given the strong link between catecholamine transmission and motivated action, we aimed to assess effect of catecholaminergic manipulation on these biases. To this end, a large sample of participants (N = 106) performed a novel motivational Go/NoGo learning task twice, once under a catecholamine challenge (methylphenidate - MPH) and once on placebo. Based on previous literature of dopaminergic drug effects (*Cools and D'Esposito, 2011*; *Frank and Fossella, 2011* for reviews), we hypothesized that MPH effects on motivated action would covary with measures scaling with baseline dopamine function, namely working memory span (*Cools et al., 2008*) and trait impulsivity (*Buckholtz et al., 2010*). Our findings are threefold: First, cue valence elicits behavioural activation in a Pavlovian manner, whereas outcome valence biases the learning of action vs. inhibition (*Figure 1A,B*). Second, MPH modulates Pavlovian biasing, while also altering the reward-driven diffusion of credit assignment during instrumental learning. Third, the direction of the effect of MPH covaries with individual differences in working memory span, but not trait impulsivity.

### Dissociable effects of cue and outcome valence on behavioural activation and instrumental learning

Cue valence affected activation versus inhibition of behaviour, consistent with previous reports (*Geurts et al., 2013*; *Guitart-Masip et al., 2012*). Even though cue valence was orthogonal to what subjects *should* be doing, subjects made more Go responses when pursuing reward, and fewer Go responses when trying to avoid punishment. We and others have previously suggested that this motivational asymmetry in behavioural activation entails Pavlovian control over instrumental behaviour (*Cavanagh et al., 2013*; *Geurts et al., 2013*; *Huys et al., 2011*). Here we challenge this initial idea, and argue that motivational valence may also bias instrumental learning. To disentangle the hypothesised contribution of a Pavlovian response bias from biased instrumental learning, we extended existing paradigms by incorporating multiple Go response options. For the cues requiring active responses, only one response option was considered correct, enabling us to disentangle general activation from specific action learning. For cues where subjects had to activate responding ('Go' cues), they increased both correct and incorrect Go responses when pursuing reward compared with when avoiding punishment. Thus, the increased activation towards reward was in part beneficial, and in part detrimental.

We used computational models to formalise our hypothesis regarding a dissociable contribution of Pavlovian activation and biased instrumental learning. We then fitted competing models to the subjects' choices, and compared the performance of all models. We demonstrate that cue valence shapes behavioural activation/inhibition in a Pavlovian manner, and additionally that outcome valence biases instrumental learning of activation/inhibition: reward enhances the learning of specific active actions, and punishment suppresses the unlearning of inactions. In short, we are quicker to believe that an action led to a reward, but reluctant to attribute a punishment to having held back.

Current views of striatal dopamine function (*Collins and Frank, 2015*; *2014*; *Frank, 2006*; *Frank, 2005*; *Lloyd and Dayan, 2016*) suggest that the striatal architecture is well suited to implement the Pavlovian asymmetry in behavioural activation. Appetitive (aversive) conditioned cues elicit peaks (dips) in mesolimbic dopamine release in the striatum (*Cohen et al., 2012*; *Day et al., 2007*; *Matsumoto and Hikosaka, 2009*; *Tobler et al., 2005*). Increased striatal dopamine levels activate the direct D1 ('Go') pathway (*Hernández-López et al., 1997*), which promotes behavioural activation (*DeLong and Wichmann, 2007*; *Mink and Thach, 1991*), whereas decreased striatal dopamine levels activate the indirect D2 ('NoGo') pathway (*Hernandez-Lopez et al., 2000*), promoting behavioural inhibition. In striatal dopamine models, increased dopamine biases action selection to be driven

more by the potential rewards of alternative actions encoded in D1 neurons and less by the costs encoded in D2 neurons (*Collins and Frank, 2014*; see also recent optogenetic experiment supporting this notion; *Zalocusky et al., 2016*), but this can also be manifest in terms of Pavlovian biases. Taken together, the striatal (in)direct pathways provide a neural mechanism for implementing Pavlovian activation to appetitive vs. aversive cues.

In parallel with our behavioural findings, the same striatal pathways may also generate the asymmetry in action learning. Here, dopamine bursts elicited by reward prediction errors (*Montague et al., 2004*; *Schultz et al., 1998*; *Schultz et al., 1997*) during the outcome, enhance long-term potentiation (LTP) of the corticostriatal synapses associated with the just-performed response (*Frank et al., 2004*). Importantly, enhancing LTP in the 'Go' pathway should promote learning of active responses, relative to learning the inhibition of actions. Recent experiments show temporally and spatially selective enhancement of corticostriatal spines given glutamatergic input (putatively representing the selected action) and followed closely in time by dopaminergic bursts (*Yagishita et al., 2014*). Thus, prolonged release of DA (e.g. after DAT blockade) might reduce this selectivity, and diffuse the specificity of credit assignment. Conversely, striatal dopamine dips following negative prediction errors can drive avoidance by promoting long-term depression (LTD) in the 'Go' pathway and LTP in the 'NoGo' pathway (*Beeler et al., 2012*; *Frank, 2005*; *Shen et al., 2008*). Indeed, transient optogenetic inhibition of DA induces behavioural avoidance of recently selected actions (*Danjo et al., 2014*; *Hamid et al., 2016*), an effect that depends on D2 receptors (*Danjo et al., 2014*). D2 neurons are excited in response to losses (*Zalocusky et al., 2016*); their activation during losses induces subsequent avoidance learning (*Kravitz et al., 2012*; *Zalocusky et al., 2016*), and their disruption prevents avoidance learning (*Hikida et al., 2010*). While LTP in the NoGo pathway would be beneficial for unlearning to perform actions, LTP in the NoGo pathway would be detrimental in case of unlearning to make NoGo responses (i.e. attributing a punishment to a NoGo response). To summarize, the dopamine peaks following positive reinforcement can enhance learning of actions by enhancing LTP in the striatal 'Go' pathway. Conversely, the dopamine dips following negative outcomes can disrupt learning to initiate responses by increasing LTD in the 'Go' pathway and LTP in the NoGo pathway.

## Methylphenidate modulates Pavlovian activation and spreads credit assignment of rewarded actions

Blocking the reuptake of catecholamines with MPH altered the extent to which subjects were influenced by the cue and outcome valence. This effect of MPH was highly variable between individuals, and depended on working memory span. In high relative to low span subjects, MPH enhanced the influence of valence, such that subjects made even more active responses when pursuing reward and displayed more inhibition when avoiding punishment. This effect was driven particularly by changes in the proportion of incorrect Go responses that subjects made. Formal modelling showed that this effect was due to MPH affecting both generalized Pavlovian activation and a diffusion of credit assignment. Specifically, MPH induced a spread of credit assignment following rewarded active responses, rather than magnifying the selective instrumental learning bias.

We argue that both of these effects can be understood as reflecting prolonged catecholamine presence in the synaptic cleft with MPH. Blocking catecholamine reuptake with MPH extends the duration of dopamine presence in the synaptic cleft (*Dreyer and Hounsgaard, 2013*). This prolonged dopamine presence (i.e. reduced temporal specificity) would be less selective in potentiating the actions that were selected immediately prior to rewards (e.g. *Yagishita et al., 2014*). This would reduce credit assignment of *specific* active actions, but still bias reinforcement of actions more generally (e.g. *Collins and Frank, 2015*; *Syed et al., 2016*). This account explains why MPH modulates the strength of the Pavlovian activation (which is inherently global) but not of the specific instrumental learning bias (which is inherently selective). Our results indeed provided evidence for this diffusing effect of MPH on the instrumental learning bias, such that reward potentiates actions globally. The data were best explained by a combination of this diffuse instrumental learning and Pavlovian response bias modulation. Thus, on the one hand MPH modulated the impact of the cue valence on behavioural activation, which surfaces already before any learning has taken place. On the other hand, MPH spread credit assignment following rewarded responses to all Go responses, which is an experience-dependent effect.

Our results are highly consistent with those predicted from current models of dopamine in the basal ganglia, suggesting that the effects of MPH are due to modulation of striatal dopamine. Of course, the present study does not allow us to exclude the possibility that (part of) the effects were mediated by extra-striatal, e.g. prefrontal regions (*Spencer et al., 2015*), or by the noradrenaline system (*Arnsten and Dudley, 2005*). Future studies are needed to investigate directly the site of the presently observed effects of MPH, e.g. with fMRI, and dopamine dependence and selectivity, e.g. with selective dopamine antagonists.

## MPH effects predicted by individual differences in working memory span

Individuals vary strongly in the extent to which MPH increases extracellular dopamine (*Volkow et al., 2002*). We therefore anticipated that the effect of MPH would covary with measures relating to baseline dopamine function. We assessed whether MPH effects were predicted by (i) working memory span, given its known relation to dopamine synthesis capacity (*Cools et al., 2008*; *Landau et al., 2009*), and (ii) trait impulsivity, for its known relation to D2 (auto)receptor availability (*Buckholtz et al., 2010*; *Kim et al., 2014*; *Lee et al., 2009*; *Reeves et al., 2012*). MPH affected choice behaviour proportional to working memory span, but not trait impulsivity. Subjects with higher working memory span, linked to higher striatal synthesis capacity, showed a relative increase in both Pavlovian response bias and spread of credit assignment under MPH. This finding that transporter blockade has stronger effects in those individuals with putatively higher baseline dopamine is in line with the observation that MPH increases dopamine levels more in individuals with higher dopamine cell activity (*van der Schaaf et al., 2013*; *Volkow et al., 2002*). Indeed, baseline dopamine cell activity is a better predictor of effects of MPH than either D2 auto-receptor availability or DAT occupancy under MPH (*Volkow et al., 2002*). Together this may explain why the observed MPH effects covary with working memory span but not trait impulsivity.

The finding that drug effects depend on working memory is highly consistent with the hypothesis that they reflect modulation of striatal dopamine (c.f. *Frank and Fossella, 2011*). However, we need to be cautious in our interpretation. First, both striatal and prefrontal dopamine are known to contribute to working memory performance (updating and maintenance respectively; e.g. *Cools and D'Esposito, 2011*). The Listening Span task does not dissociate between working memory updating and maintenance, and thus a contribution of modulation of prefrontal dopamine cannot be excluded. Another possibility raised by the finding that drug effects depend on span, is that they reflect modulation of working memory itself, rather than reflecting dependence on baseline dopamine synthesis capacity. However, we argue that this is unlikely, because there was no significant effect of baseline working memory on motivational bias under placebo conditions. Rather, this relationship was induced by MPH. For future studies, it would be of interest to also include other measures related to baseline dopamine levels, such as eyeblink rates. More broadly, further research is required to identify the optimal combination of the various proxy measures of individual variability in the dopamine system in order to account for the large inter-individual variability in dopaminergic drug response. This is one of the major aims of our ongoing work.

Across subjects, MPH increased subjective experiences of positive affect and alertness, and decreased calmness (Appendix 2). In contrast to the MPH-induced Pavlovian response bias and diffuse learning bias, these non-specific mood changes did not covary with working memory span. In other words, the MPH-induced mood changes are orthogonal to our effect of interest. Therefore, the MPH effect on Pavlovian activation and biased instrumental learning cannot be attributed to MPH-induced changes in mood.

## Conclusion

This study elucidates two distinct mechanisms by which motivational valence can bias behaviour. Cue valence promotes activation/inhibition in a Pavlovian manner, whereas outcome valence affects action/inhibition learning. Blocking the reuptake of catecholamines with methylphenidate altered the Pavlovian response bias, and had a diffuse, rather than selective, effect on biased learning. The effect of methylphenidate on the Pavlovian bias and biased learning was predicted by working memory span, such that methylphenidate enhanced Pavlovian activation and biased learning proportional

to working memory span. These results help bridge the study of motivational biasing of action and instrumental learning, and help refine current models of catecholamines in motivated action.

The present observations suggest that we need to add a new dimension to the suggested dichotomy of the role of dopamine in learning versus performance. Our study brings together two literatures that emphasise the role of (midbrain) dopamine in reward (prediction-error) based learning on the one hand (*Collins and Frank, 2014*; *Frank et al., 2004*; *Schultz et al., 1997*), and motivation-driven performance and behavioural activation on the other (*Beierholm et al., 2013*; *Berridge, 2007*; *Robbins and Everitt, 2007*; *Shiner et al., 2012*; *Smittenaar et al., 2012*). Our results suggest that these two interact, resulting in biased learning of action-reward and inaction-punishment links, putatively via the same striatal mechanism that drive motivational Pavlovian response biases. Like motivational response tendencies, such biased learning would allow us to optimally profit from stable environmental statistics, as this instrumental learning bias supports rapid learning of likely action-outcome associations (e.g. that an action caused a reward), while avoiding learning unlikely, spurious, associations (e.g. that inhibition caused a punishment).

## Materials and methods

### General procedure and pharmacological manipulation

The study consisted of two test sessions with an interval of one week to two months. The first test day started with informed consent, followed by a medical screening. Participation was discontinued if subjects met any of the exclusion criteria (Appendix 1). On both test days, subjects first completed baseline measures. Next subjects received a capsule containing either 20 mg MPH (Ritalin, Novartis) or placebo, in a double-blind, placebo-controlled, cross-over design. MPH blocks the dopamine and noradrenaline transporters, thereby diminishing the reuptake of catecholamines. When administered orally, MPH has a maximal plasma concentration after 2 hr and a plasma half-life of 2–3 hr (*Kimko et al., 1999*). After an interval of 50 min, subjects started with the task battery containing the motivational Go/NoGo learning task. See Appendix 2 for an overview of the task battery. On average the motivational Go/NoGo learning task was performed 2 hr after capsule intake, well within the peak of plasma concentration. Both test days lasted approximately 4.5 hr, which subjects started at the same time (maximum difference of 45 min). Blood pressure, mood and potential medical symptoms were monitored three times each day: before capsule intake, upon start of the task battery and after finishing the task battery. Subjects were told to abstain from alcohol and recreational drugs 24 hr prior to testing and from smoking and drinking coffee on the days of testing. Subjects completed self-report questionnaires at home between (but not on) test days. Upon completion of the study, subjects received a monetary reimbursement or study credits for participation. The study was in line with the local ethical guidelines approved by the local ethics committee (CMO / METC Arnhem Nijmegen: protocol NL47166.091.13), pre-registered (trial register NTR4653, http://www.trialregister.nl/trialreg/admin/rctview.asp?TC=4653), and in accordance with the Helsinki Declaration of 1975. Baseline measures, self-report questionnaires, mood- and medical symptom-ratings are reported in Appendix 2.

### Subjects

As individual differences were a main focus of the study, we collected a large sample of 106 native Dutch volunteers (aged 18–28 years, mean (SD) = 21.5 (2.3); 53 women; 84 right-handed; sample size calculation reported in CMO protocol NL47166.091.13). Four subjects dropped out after the first test day (due to too much delay between test days, loss of motivation, nausea, and mild arrhythmia). Two subjects dissolved the capsules before swallowing and are discarded because of uncertainty in the pharmacodynamics. One subject did not sufficiently engage in the task (only 13/2% Go responses on day 1/2) and was discarded as well. We repeated the analyses with these subjects included to confirm that this did not alter the conclusions (Appendix 3). Of the resulting 99 subjects, 48 subjects received MPH on the first day. Exclusion criteria comprised a history of psychiatric, neurological or endocrine disorders. Appendix 1 presents a complete overview of the exclusion criteria.

## Motivational Go/NoGo learning task

Each trial started with the on-screen presentation of a cue (*Figure 2A*). During cue presentation subjects could decide to press a button (*Go response*) or not (*NoGo response*). Subjects could either press the left (*Go-left*) or right (*Go-right*) button on a button box. Subjects received feedback based on their response.

Each cue had a red or green edge. Cues with a red edge (*Avoid cues*) were followed by neutral feedback or punishment. Cues with a green edge (*Win cues*) were followed by reward or neutral feedback. Subjects were informed about these contingencies. Note that the explicit cue valence is in contrast to previous studies where subjects needed to learn the cue valence during the task (e.g. *Cavanagh et al., 2013*; *Guitart-Masip et al., 2012*). The rationale of explicit cue valence was to directly observe effects of cue valence on choice and minimize individual differences in learning the cue valence. Punishment consisted of the display of the red text '−100', accompanied by a low buzz, reward of the green text '+100' together with a flourish sound, and the neutral feedback of the grey text '000' together with a short beep. All cues had unique shapes and colours well distinguishable from the red and green edge. Cue colour and shape were randomized over cue types. Two separate stimulus sets were used for the two test days to prevent transfer effects, and set order was counterbalanced across subjects.

For each cue, there was one correct response (Go-left, Go-right or NoGo; *Figure 2C*), which subjects had to learn by trial and error. Feedback validity was 80%, that is, correct (incorrect) responses were followed by the desirable outcome 80% (20%) of the time (*Figure 2D*). There were eight cues in total (*Figure 2B*). The number of Go and NoGo cues was kept equal to prevent reinforcing an overall Go bias.

The order of cue presentation was pseudorandom, as cues could be repeated once at most. Each cue was presented 40 times. The task lasted approximately 30 min, including instructions and a self-paced break halfway. The instructions were presented on screen. Subjects were informed about the probabilistic nature of the feedback and that each cue had one optimal response. At the end of the task the total number of points won or lost was displayed on screen and subjects were informed beforehand that these points would be converted to a monetary bonus at the end of the study (mean = EUR2.90, SD = 1.49).

## Listening span test

Working memory span was assessed with the Listening Span Test (*Daneman and Carpenter, 1980*; *Salthouse and Babcock, 1991*), which was also used in two FMT PET studies showing positive correlations with striatal dopamine synthesis capacity (*Cools et al., 2008*; *Landau et al., 2009*). Subjects completed the Listening Span Test on day two prior to capsule intake. The Listening Span Test consists of sets of pre-recorded sentences, increasing from 2 to 7 sentences. Subjects are presented with the sentences, and required to simultaneously answer written verification questions regarding the content of each sentence. At the end of each set, subjects recalled the final word of each sentence in the order of presentation. The Listening Span reflects the set size of which the subject correctly recalled the final words on at least two out of three trials. Listening span increased with half a point, when only one trial of the next level was correct.

## Barratt impulsiveness scale

Trait impulsivity was assessed with the Barratt Impulsiveness Scale (BIS-11) (*Patton et al., 1995*). The BIS-11 is a self-report questionnaire, consisting of 30 questions tapping in common (non)impulsive behaviours and preferences. The BIS-11 total impulsivity scores reflect the tendency towards impulsivity. Subjects completed the questionnaire at home between test days.

## Statistical analyses

To assess the influence of motivational valence on behavioural activation, we first analysed Go vs. NoGo responses (irrespective of Go-left vs. Go-right). Second we tested whether effects on Go responses were explained by correct or incorrect Go responses. We were specifically interested how MPH altered Go/NoGo responding to Win vs. Avoid cues as a function of Listening Span and Impulsivity.

To account for both between and within subject variability, choice data were analysed with logistic mixed-level models using the lme4 package in R (**Bates et al., 2014**; **R Developement Core Team, 2015**). Reflecting our objectives, the mixed models included the within subject factors Drug (MPH vs. placebo), Valence (Win vs. Avoid cue), and Required Action (Go vs. NoGo), and the between subject factors Listening Span and Impulsivity. The analysis of correct and incorrect Go responses included only the Go cues; hence this analysis did not include the factor Required Action. Models included all main effects and interactions, except for the interactions between Listening Span and Impulsivity. All models contained a full random effects structure (**Barr, 2013**; **Barr et al., 2013**). We performed control analyses using a model comparison approach, where we tested whether the following factors improved model fit: Drug Order, Testing Day, Gender, and NLV (a measure for verbal intelligence). For completeness, we analysed reaction times (RTs) as a measure of behavioural vigour (Appendix 4).

## Computational modelling – Pavlovian response bias and instrumental learning bias

In all models, action weights ($w$) are estimated for each response option ($a$) for all trials ($t$) per cue ($s$). Based on these action weights choice probabilities are computed using a softmax function, as follows:

$$p(a_t|s_t) = \left[ \frac{\exp(w(a_t,s_t))}{\sum_{a'} \exp(w(a',s_t))} \right] \tag{1}$$

In the simplest model (M1) the action weights are fully determined by the learned action values ($Q$-values). To compute the action values, we used standard delta-rule learning with two free parameters; a learning rate ($\varepsilon$) scaling the update term, and feedback sensitivity ($\rho$) scaling the outcome value (comparable to the softmax temperature).

$$Q_t(a_t,s_t) = Q_{t-1}(a_t,s_t) + \varepsilon(\rho r_t - Q_{t-1}(a_t,s_t)) \tag{2}$$

Here outcomes are reflected by $r$, where $r \in \{-1,0,1\}$. In the current paradigm cue valence is instructed, by means of the green and red cue edges. Therefore, the initial expected outcome is 0.5 for Win cues and $-0.5$ for Avoid cues. Initial $Q$-values ($Q_0$) are set accordingly to $\rho*0.5$ for Win cues and $\rho*-0.5$ for Avoid cues.

In M2 a go bias parameter ($b$) is added to the action weights of Go responses. We then explored the influence of Pavlovian biases that modulate Go responding according to predicted reward value. Pavlovian values ($V$) contribute to the action weights in M3a, increasing (decreasing) the weight of Go responses for positive (negative) Pavlovian values respectively.

$$w(a_t,s_t) = \begin{cases} Q(a_t,s_t) + \pi V(s) + b & if \ a = Go \\ Q(a_t,s_t) & else \end{cases} \tag{3}$$

Here the weight of the Pavlovian values is determined by the parameter $\pi$. Pavlovian values are fixed at 0.5 for Win cues and at $-0.5$ for Avoid cues, again because cue valence is instructed.

In M3b we included the instrumental learning bias parameter ($\kappa$) instead of the Pavlovian bias, to assess whether the motivational bias can be explained in terms of enhanced learning of Go following a reward, and disrupted learning from punishment following NoGo.

$$\epsilon = \begin{cases} \epsilon_0 + \kappa & if \ r_t = 1 \ \& \ a = go \\ \epsilon_0 - \kappa & if \ r_t = -1 \ \& \ a = nogo \\ \epsilon_0 & else \end{cases} \tag{4}$$

In model M4, we included both the Pavlovian bias parameter and the instrumental learning bias parameter.

We used a sampling method for hierarchical Bayesian estimation of group-level and subject-level parameters. The group-level parameters ($X$) serve as priors for the individual-level parameters ($x$), such that $x \sim \mathcal{N}(X,\sigma)$. The hyperpriors for $\sigma$ are specified by a half-Cauchy (**Gelman, 2006**) with a scale of 2. The hyperpriors for $X$ are centered around 0 (with the exception of $X_\rho$) and weakly informative: $X_\rho \sim \mathcal{N}(2,3)$, $X_{\varepsilon,\kappa} \sim \mathcal{N}(0,2)$, $X_{b,\pi} \sim \mathcal{N}(0,3)$. All parameters are unconstrained, with the

exception of $\rho$ (positivity constraint; exponential transform) and $\varepsilon$ ([0 1] constraint; inverse logit transform). To ensure that the effect of $\kappa$ on $\varepsilon$ (*Equation 4*) was symmetrical in model space (i.e. after sigmoid transformation to ensure [0 1] constraint), $\varepsilon$ was computed as:

$$\varepsilon = \begin{cases} \varepsilon_0 & = \mathrm{inv.logit}(\varepsilon) \\ \varepsilon_{\mathrm{punished\,NoGo}} & = \mathrm{inv.logit}(\varepsilon - \boldsymbol{\kappa}) \\ \varepsilon_{\mathrm{rewarded\,Go}} & = \varepsilon_0 + \left(\varepsilon_0 - \varepsilon_{\mathrm{punished\,NoGo}}\right) \end{cases} \tag{5}$$

Model estimations were performed using Stan software in R (RStan) (*Stan Development Team, 2016*). Stan provides full Bayesian inference with Markov chain Monte Carlo (MCMC) sampling methods (*Metropolis et al., 1953*). The number of Markov chains was set at 4, with 200 burn-in iterations and 1000 post burn-in iterations per chains (4000 total). Model convergence was considered when the potential scale reduction factor $\hat{R} < 1.1$ for all parameters (*Gelman and Rubin, 1992*). In case model convergence was not reached, both (post) burn-in samples were increased to 1500. Not all models reached convergence at this point. Therefore, we repeated model estimation while excluding the subjects (N = 5) for whom initially $\hat{R} > 1.1$ in any one of the models, resulting in model convergence for all models. We report model evidence including all subjects in Appendix 5, showing that model selection and parameter inference remains the same when excluding these subjects. Model comparison was evaluated using the Watanabe-Akaike Information Criteria (WAIC) (*Watanabe, 2010*). WAIC is an estimate of the likelihood of the data given the model parameters, penalized for the effective number of parameters to adjust for overfitting. Lower (i.e. more negative) WAIC values indicate better model fit. As WAIC is reported on the deviance scale (*Gelman et al., 2014*), a difference in WAIC value of 2–6 is considered positive evidence, 6–10 strong evidence, and >10 very strong evidence (*Kass and Raftery, 1995*).

## Computational modelling – Effects of methylphenidate

Having established the mechanisms by which motivational valence may affect instrumental learning and activation, we extended the winning model to test which of these mechanisms are affected by MPH, putatively driven by a prolonged striatal presence of catecholamines (dopamine) following reward, due to reuptake inhibition by MPH.

In M5 we tested whether MPH altered the Pavlovian response bias. This model includes a parameter allowing for an MPH-induced change in the Pavlovian weight ($\pi_{\mathrm{MPH}}$):

$$\pi = \begin{cases} \pi_0 & \textit{if placebo} \\ \pi_0 + \pi_{MPH} & \textit{if MPH} \end{cases} \tag{6}$$

Next, we tested two mechanisms by which MPH might alter the bias in instrumental learning ($\kappa$). In M5b we tested whether MPH simply enhanced or reduced the learning bias parameter, estimating an additive effect of $\kappa_{\mathrm{MPH\text{-}selective}}$:

$$\kappa = \begin{cases} \kappa_0 & \textit{if placebo} \\ \kappa_0 + \kappa_{\mathrm{MPH-selective}} & \textit{if MPH} \end{cases} \tag{7}$$

Alternatively, the prolonged presence of catecholamines following reward under MPH could induce a more diffuse credit assignment, rather than a selective learning bias effect. To test this hypothesis, in M5c we included a MPH-induced learning bias parameter ($\kappa_{\mathrm{MPH\text{-}diffuse}}$), which was used to update *both* Go responses, on all trials where any active Go response was followed by reward, in addition to the regular learning update for the chosen Go response:

$$\textit{if } MPH, r_t = 1, \& \, a_{chosen} = Go:$$
$$Q_t\left(a_{chosenGo,t}, s_t\right) = Q_{t-1}\left(a_{chosenGo,t}, s_t\right) + \left(\varepsilon + \kappa_0 + \kappa_{\mathrm{MPH-diffuse}}\right) \cdot PE \tag{8}$$
$$Q_t\left(a_{unchosenGo,t}, s_t\right) = Q_{t-1}\left(a_{unchosenGo,t}, s_t\right) + \kappa_{\mathrm{MPH-diffuse}} \cdot PE$$

Where $PE$ is the prediction error following the rewarded Go response: $PE = \rho r_t - Q_{t-1}(a_t, s_t)$. Thus where $\kappa_{\mathrm{MPH\text{-}selective}}$ enhances the learning of the selected Go response after reward, $\kappa_{\mathrm{MPH\text{-}diffuse}}$ induces learning of all Go responses when a Go response elicited reward.

To test whether MPH affected both the Pavlovian response bias and instrumental learning bias, M6 include $\pi_{MPH}$ parameter as well as the winning model of the two learning bias mechanisms (M5c - $\kappa_{MPH\text{-}diffuse}$). For completeness, we report the composite model including the parameters $\pi_{MPH}$ and $\kappa_{MPH\text{-}selective}$ in Appendix 5. The hyperpriors are again centered around 0 and weakly informative: $X_{\kappa mph} \sim \mathcal{N}(0,2)$ and $X_{\pi mph} \sim \mathcal{N}(0,3)$, where only $X_{\kappa mph\text{-}diffuse}$ is constrained ([0 1] constraint; inverse logit transform).

Having established the winning model, we used two absolute model fit approaches to confirm that the winning model captures the effects of interest; the post-hoc absolute-fit approach (also called *one-step-ahead prediction*) and posterior predictive model simulation approach (*Steingroever and Wagenmakers, 2014*). The posterior predictive model simulations simply 'play' the task, using the estimated parameters. This approach, however, ignores sequential/history effects of actually observed choices and outcomes. The 'one-step-ahead' prediction fits parameters to trials $t_1$ - $t_{n-1}$, and then predicts the choice on trial $t_n$. Taking these sequential effects into account is particularly important to assess effects of the parameters that estimate the effect of previous choice/outcome combinations, i.e. the learning rate parameters, relative to the constant parameters like the Pavlovian and go biases. For both the one-step-ahead predictions and model simulations, we computed action probabilities for all subjects on all trials using the sampled combinations of all individual-level parameter estimates. For the one-step-ahead predictions the observed choices and outcomes were used to update the action probabilities. For the model simulations choices were simulated depending on the computed action probabilities, and outcomes were determined according to the ground-truth outcome probabilities (i.e. a correct response would lead to the desired outcome 80% of the time). Subsequently, outcomes corresponding to the simulated choices were used to update the action probabilities. The one-step-ahead prediction and simulations were repeated for all sampled parameter combinations (4000 times), and action probabilities were averaged over repetitions. Averaging over repetitions also minimizes effects of randomness due to the stochastic nature of the choice simulation.

## Acknowledgements

We thank Monique Timmer and Peter Mulders for medical assistance; Dr. Sean James Fallon for advice on setting up the MPH study; Dr. Matthew Albrecht for help with RStan.

## Additional information

### Competing interests

MJF: Reviewing editor, *eLife*. HEMdO: HEMdO has acted as a consultant for Eleusis benefit corps, but does not own shares. The other authors declare that no competing interests exist.

### Funding

| Funder | Grant reference number | Author |
|---|---|---|
| Nederlandse Organisatie voor Wetenschappelijk Onderzoek | Research talent grant,406-14-028 | Jennifer C Swart |
| University of Birmingham | Birmingham Fellows Programme | Jennifer L Cook |
| ZonMw | 92003576 | Dirk EM Geurts |
| National Science Foundation | 1460604 | Michael J Frank |
| National Institute of Mental Health | R01 MH080066-01 | Michael J Frank |
| James S. McDonnell Foundation | James McDonnell scholar award | Roshan Cools |
| Nederlandse Organisatie voor Wetenschappelijk Onderzoek | Vici Award,453.14.005 | Roshan Cools |
| Nederlandse Organisatie voor Wetenschappelijk Onderzoek | Veni grant,451-11-004 | Hanneke EM den Ouden |

The funders had no role in study design, data collection and interpretation, or the decision to submit the work for publication.

## Author contributions

JCS, Designed the paradigm, Set up the MPH study, Collected the data, Analysed the data, Developed the computational models, Wrote and revised the manuscript; MIF, JLC, Set up the MPH study, Collected the data, Edited / revised the manuscript; DEMG, Set up the MPH study, Edited / revised the manuscript; MJF, Designed the paradigm, Developed the computational models, Revised / edited the manuscript; RC, Designed the paradigm, Set up the MPH study, Revised / edited the manuscript; HEMdO, Designed the paradigm, Set up the MPH study, Collected the data, Analysed the data, Developed the computational models, Wrote / revised the manuscript

## Author ORCIDs

Jennifer C Swart, http://orcid.org/0000-0003-0989-332X
Dirk EM Geurts, http://orcid.org/0000-0002-2505-1301
Michael J Frank, http://orcid.org/0000-0001-8451-0523

## Ethics

Human subjects: Informed consent, and consent to publish, was obtained prior to participation. The study was in line with the local ethical guidelines approved by the local ethics committee (CMO / METC Arnhem Nijmegen: protocol NL47166.091.13), pre-registered (trial register NTR4653, http://www.trialregister.nl/trialreg/admin/rctview.asp?TC=4653), and in accordance with the Helsinki Declaration of 1975.

# Additional files

## Supplementary files

• Source code 1. Source code for behavioural analysis. This zip-file contains source code for (1) deblinding, (2) descriptives & demographics, (3) mood ratings, (4) motivational Go-NoGo task, and a README.txt file with description of the source code and code structure.

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

## Appendix 1

### Exclusion criteria

Exclusion criteria comprised a history of psychiatric, neurological or endocrine disorders. Further exclusion criteria were autonomic failure, hepatic, cardiac, obstructive respiratory, renal, cerebrovascular, metabolic, ocular or pulmonary disease, epilepsy, substance abuse, suicidality, hyper/hypotension, diabetes, pregnancy/breastfeeding, lactose intolerance, abnormal hearing or (uncorrected) vision (e.g. colour blindness), irregular sleep/wake rhythm, regular use of corticosteroids, use of MAO inhibitor, anaesthetic, anti-depressant or anti-psychotic drugs within the week prior to the start of the study, use of psychotropic medication or recreational drugs/alcohol 24 hr before each test day, and first degree family members with schizophrenia, bipolar disorder, ventricular arrhythmia or sudden death. Inclusion age range was 18–45 years old.

## Appendix 2

### Baseline measures and mood ratings

Prior to capsule intake, subjects completed a Dutch reading test (NLV, *Schmand et al., 1991*) as a proxy of verbal intelligence on day 1, and the Listening Span Test (*Daneman and Carpenter, 1980*; *Salthouse and Babcock, 1991*) on day 2. Subsequently subjects completed the Digit Span Test (forward and backward; *Wechsler, 2008*) and the training phase of a Pavlovian-Instrumental Transfer task (PIT, *Geurts et al., 2013*; *Huys et al., 2011*) of which data will be reported elsewhere. Between test days, subjects completed a number of self-report questionnaires. The group that received MPH on day 1 did not differ significantly on any of the baseline measures from the group that received placebo on day 1 (p<0.05). See *Appendix 2—table 1* for an overview of the neuropsychological test scores and self-report questionnaires.

**Appendix 2—table 1.** Mean(SD) scores for neuropsychological tests and self-report questionnaires for the group that received placebo and MPH on day 1. Significance levels for the between group differences are reported. Self-report questionnaires include the Barratt Impulsiveness Scale (BIS-11; *Patton et al., 1995*), the Behavioural Inhibition Scale/Behavioural Activation Scale (BISBAS; *Carver and White, 1994*), Need for Cognition Scale (NCS, *Cacioppo et al., 1984*), Multidimensional Scale of Perceived Social Support (MSPSS, *Zimet et al., 1988*), Barratt Simplified Measure of Social Status (BSMSS, *Barratt, 2006*), Sociable and Aggressive Dominance Questionnaire (SADQ, *Kalma et al., 1993*), Beck Depression Inventory II (BDI-II; *Beck et al., 1996*), Spielberger Trait Anxiety Inventory (STAI; *Spielberger et al., 1983*).

| | | Group 1 Placebo Day 1 | Group 2 MPH Day 1 | |
|---|---|---|---|---|
| *Neuropsychological tests* | Listening span | 5.0 (0.9) | 4.6 (1.2) | p=0.16 |
| | NLV | 94.4 (7.6) | 92.6 (7.6) | p=0.23 |
| | Digit span – forward | 17.2 (3.7) | 16.2 (3.6) | p=0.16 |
| | Digit Span - backward | 14.7 (3.4) | 13.9 (2.7) | p=0.22 |
| *Self-report questionnaires* | Impulsivity (BIS-11) | 63.5 (8.9) | 60.2 (7.9) | p=0.052* |
| | Behavioural inhibition (BIS) | 16.4 (3.7) | 16.3 (3.5) | p=0.90 |
| | Behavioural activation (BAS) | 22.8 (3.9) | 23.9 (4.0) | p=0.17 |
| | Need for cognition (NCS) | 64.5 (10.5) | 62.2 (10.5) | p=0.26 |
| | Social support (MSPSS) | 71.1 (10.1) | 69.3 (9.6) | p=0.35 |
| | Social status (BSMSS) | 49.8 (12.1) | 45.9 (12.7) | p=0.11 |
| | Social dominance (SADQ) | 4.1 (0.9) | 4.1 (0.8) | p=0.82 |
| | Aggressive dominance (SADQ) | 2.6 (0.6) | 2.6 (0.6) | p=0.69 |
| | Depressive symptoms (BDI-II) | 3.5 (3.7) | 3.6 (3.9) | p=0.97 |
| | Anxiety symptoms (STAI) | 32.4 (6.6) | 32.4 (7.2) | p=1.0 |

*One subject had an outlying score on the BIS-11. Without outlier: p=0.09.

Mood ratings, heart rate and blood pressure were monitored for safety reasons three times during each test day, (i) before capsule intake, (ii) upon start task battery, and (iii) upon completion of the task battery. The mood ratings consisted of the Positive and Negative Affect Scale (PANAS, *Watson et al., 1988*) and the Bond and Lader Visual Analogues Scales (calmness, contentedness, alertness; *Bond and Lader, 1974*), as well as a medical Visual Analogues Scale.

We assessed whether MPH affected mood and medical symptoms. For this control analysis we performed a repeated measures MANOVA using Pillai's trace with the within subject

factors Time (baseline/start testing/end testing) and Drug (MPH/placebo), and dependent variables Positive Affect, Negative Affect, Calmness, Contentedness, Alertness, and Medical Symptoms. Significant effects were further explored with Bonferonni corrected repeated measures ANOVA, where alpha = 0.05/6 ≈ 0.008. Greenhouse-Geisser correction was applied when the assumption of sphericity was not met.

MPH affected these self-report ratings (Time x Drug: $V = 0.38$, $F(12,90) = 4.7$, p<0.001), in the absence of baseline differences between the MPH and placebo groups ($V = 0.07$, $F(6,96) = 1.1$, p=0.359). After capsule intake MPH increased Positive Affect ($F(1,101) = 17.5$, p<0.001), Alertness ($F(1,101) = 15.2$, p<0.001), and Medical Symptoms ($F(1,101) = 11.1$, p=0.001), and decreased Calmness ($F(1,101) = 8.6$, p=0.004), relative to placebo. We confirmed that the effects of MPH on the self-report ratings did not further interact with Listening Span and Impulsivity (p>0.05). Thus, the MPH-induced changes in mood and medical symptoms were orthogonal to the Listening Span dependent MPH effects we observed in the task.

# Appendix 3

## General performance and individual differences in drug effects on task performance

In the main manuscript, we report the results of 99 (out of 106) subjects. Four subjects did not complete both test days, two subjects dissolved the capsules before swallowing, and one subject did not sufficiently engage in the task (only 13/2% Go responses on day 1/2). We then repeated the analyses with these subjects included to confirm that this did not alter the conclusions (*Appendix 3—figure 1*).

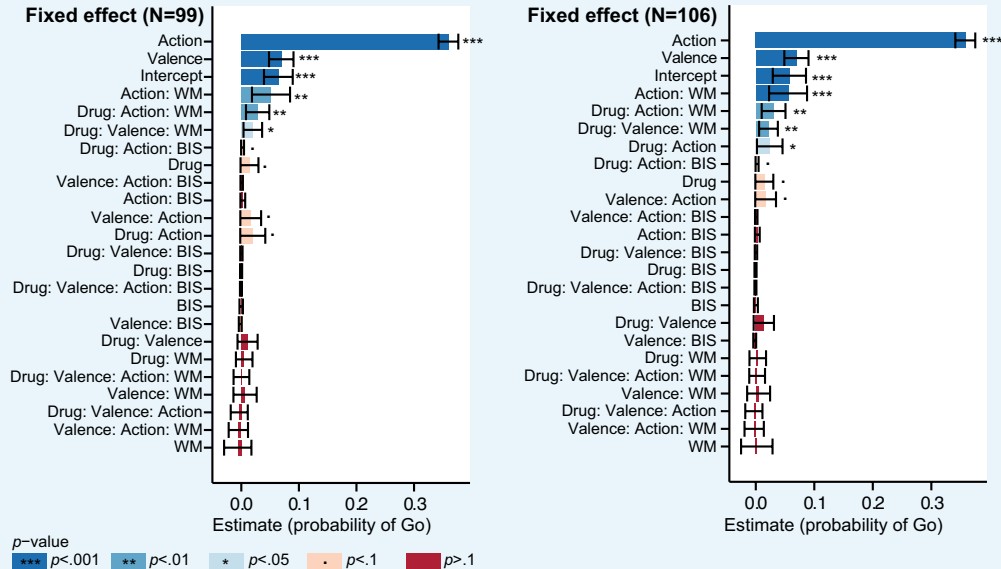

**Appendix 3—figure 1.** Logistic mixed model estimates of the probability of Go responses to verify that exclusion of a subset of subjects (7) did not affect our inference. Left: N = 99; Right: N = 106. Fixed effect estimates and 95% confidence interval (CI) are plotted on probability scale. Effects are sorted by lower bound of the CI. The results including all 106 subjects replicate the findings when discarding the subset of subjects (four subjects dropped out after the first test day, two subjects dissolved the capsules before swallowing, one subject did not sufficiently engage in the task).

MPH increased the proportion of Go responses to cues requiring a Go response depending on working memory span (Required Action x Drug x Listening Span: $X^2(1)=7.5$, p=0.006). Under MPH, higher span subjects made more Go responses to Go than NoGo cues (MPH: $X^2(1)=18.3$, p<0.001), while this was not the case under placebo (Placebo: $X^2(1)=1.2$, p=0.264). This effect of MPH was not significant across the group (independent of span) either (Required Action x Drug: $X^2(1)=3.2$, p=0.073). Thus, independent of the cue valence, MPH altered Go/NoGo responding as a function of the optimal action. Again, this effect of MPH covaried with working memory span, and not trait impulsivity (*Appendix 3—figure 1*). High span subjects made more (fewer) Go responses to cues requiring Go (NoGo) responses under MPH relative to placebo. Low span subjects showed the opposite pattern. These results could be interpreted as a cognitive enhancing effect of MPH in high span subjects, but not in low span subjects. This MPH-induced response accuracy is orthogonal to our effect of interest, and could thus not be attributed to an altered Pavlovian response bias or instrumental learning bias. Although this MPH effect on response accuracy is interesting in itself, it was not the focus of the current study, and therefore serves primarily as an invitation

for future studies to assess the cognitive enhancing effects of MPH on instrumental responding.

Swart *et al.* eLife 2017;6:e22169. DOI: 10.7554/eLife.22169

## Appendix 4

### Reaction times

For completeness, we analysed reaction times as a measure of behavioural vigour. First, we confirmed that the expected task effects are present. Second, we assessed whether the MPH effects on Go responding were accompanied by effects on RT, potentially indicative of a speed-accuracy trade-off. RT data were log(ln)-transformed to improve normality and analysed with linear mixed-level models using the lme4 package in R (***Bates et al., 2014***; ***R Developement Core Team, 2015***). We assessed RTs of all Go responses, irrespective of the accuracy of the Go responses, in a model including the within subject factors Drug (MPH vs. placebo), Valence (Win vs. Avoid cue), and Required Action (Go vs. NoGo), and the between subject factor Listening Span.

Regarding the expected task effects, subjects were faster when they made Go responses to Go vs. NoGo cues (Required Action: $X^2(1)=296.2$, p<0.001), indicative of learning (i.e. faster to correct than incorrect responses). We also observed effects of the motivational bias in reaction times, where cue valence influenced RTs (Valence: $X^2(1)=89.5$, p<0.001), such that RTs were shorter to Win vs. Avoid cues. This effect of cue valence was stronger for NoGo compared to Go cues (Required Action x Valence: $X^2(1)=11.5$, p<0.001), though both were highly significant (Go: $X^2(1)=53.7$, p<0.001; NoGo: $X^2(1)=66.6$, p<0.001).

Regarding the MPH effects on RT, there was no effect of MPH on the motivational valence effect on RT (Valence x Drug: $X^2(1)=0.8$, p=0.37), in line with the absence of any MPH main effect on Go responding. In contrast to Go responding, there were no Listening Span-dependent effects of MPH on RTs (all p>0.7). The absence of MPH effects on RTs suggests that the MPH effects reported in the main manuscript are not due to an effect on speed-accuracy trade-off. Perhaps of interest, but beyond the scope of this article, is that we did observe span-dependent effects independent of drug treatment. Higher span subjects sped up more for Win relative to Avoid cues (Valence x Listening Span: $X^2(1)=4.2$, p=0.041), and for Go relative to NoGo cues (Required Action x Listening Span: $X^2(1)=5.2$, p=0.023). No other effects were significant (p>0.05).

## Appendix 5

### Computational modelling

In the main article, we report five base models (M1, M2, M3a, M3b, M4) to disentangle the role of Pavlovian and instrumental learning mechanisms in driving motivational biasing of action. The winning base model was then extended in three competing models (M5a-c) and a composite model (M6) to assess the effects of MPH on these mechanisms. Not all models reached convergence when including all subjects of the behavioural analysis (N = 99). For five subjects, $\hat{R}R$ exceeded 1.1 in one or more of the models M1/M2/M5a/M6. Therefore, we repeated model estimation while excluding the five subjects for whom initially $\hat{R}R$ exceeded 1.1 in any one of the models, resulting in model convergence for all models (see main article). In *Appendix 5—figure 1A-E* we report the model comparison results and parameter inference for the models including all subjects, to demonstrate our conclusions do not depend on the exclusion of these five non-converging subjects.

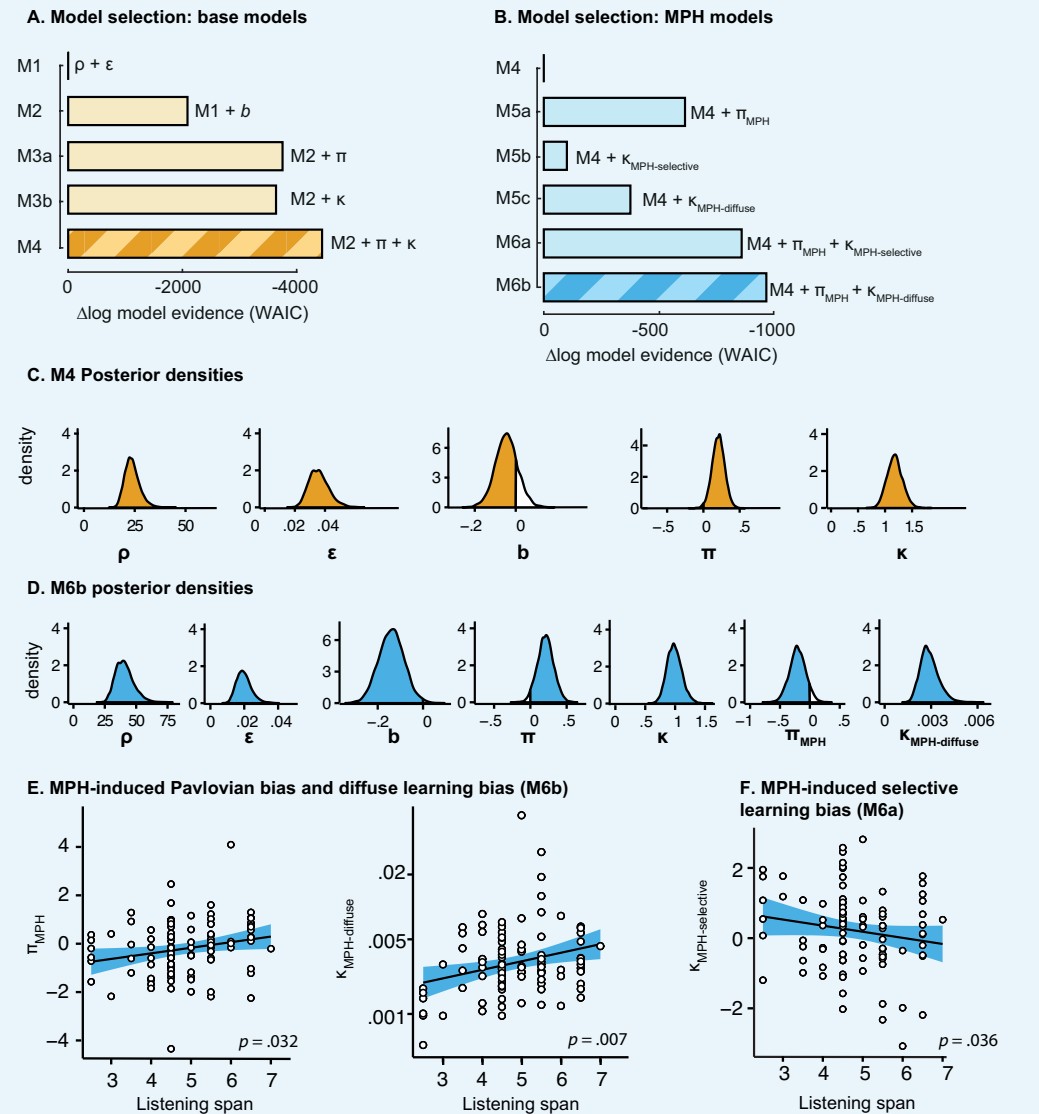

**A. Model selection: base models**

**B. Model selection: MPH models**

**C. M4 Posterior densities**

**D. M6b posterior densities**

**E. MPH-induced Pavlovian bias and diffuse learning bias (M6b)**

**F. MPH-induced selective learning bias (M6a)**

**Appendix 5—figure 1.** Model selection and parameter inference for base models and extended MPH models including all subjects (N = 99). (**A–B**) Model selection favours M4 of the base

models and M6b of the extended MPH models as reported in the main article. Note that M6b in this figure corresponds to M6 in the main manuscript. (**C-D**) Posterior densities of top-level parameters of the winning base and MPH model, in model space (i.e. transformed). Only $\kappa$ is presented untransformed (i.e. in sample space), as it is added to $\varepsilon_0$ prior to transformation. (**E**) As reported in the main article, $\pi_{MPH}$ and $\kappa_{MPH-diffuse}$ of M6b positively correlate with Listening Span. (**F**) In the composite model M6a, $\kappa_{MPH-selective}$ correlates negatively with Listening Span (N = 94; Rho = −0.22, p=0.036), which further supports that this parameter cannot capture the positive relation between listening span and the effect of MPH on motivational bias. Note that we report the correlation here for the 94 subjects for whom the parameters were reliably estimated, i.e. model convergence was reached.

Note that the go bias estimates of the winning base model M4 did not significantly deviate from 0 across the group (83% of posterior distribution <0, cut-off is usually considered 90%). The fact that inclusion of this go bias parameter did improve model evidence suggests large individual variance. In other words, inclusion of this parameter was important for explaining the data, but the direction of its effect was variable across subjects. It is noteworthy that the go bias estimates are on average negative (even if not significantly different from 0), in contrast to previous studies (*Cavanagh et al., 2013*; *Guitart-Masip et al., 2012*). This discrepancy likely is the result of incorporation of the additional Go response in the current paradigm, such that chance level of Go responses is 67%, rather than 50%, and so a positive bias estimate corresponds to a greater than 2/3 proportion of Go responses overall.

Furthermore, in the extended MPH models, both $\pi_{MPH}$ (M5a) and $\kappa_{MPH-diffuse}$ (M5c) greatly increased model evidence, in contrast to addition of $\kappa_{MPH-selective}$ (M5b), which only marginally increased model evidence. Therefore, to assess whether both Pavlovian ($\pi_{MPH}$) and instrumental learning ($\kappa_{MPH-diffuse}$) effects explained Listening Span-dependent MPH variance independently (i.e. whether there was evidence for both of these effects), we constructed a composite model (M6 in main text; M6b in *Appendix 5—figure 1*) containing both parameters. For completeness, here we also report the composite model containing both the $\pi_{MPH}$ and $\kappa_{MPH-selective}$ parameters (M6a). As expected, model selection favours the composite model with a spread of credit assignment ($\kappa_{MPH-diffuse}$, M6b; WAIC$_{N=94}$ = 66069) over the model that includes a strengthening of the selective instrumental learning bias ($\kappa_{MPH-selective}$, M6a; WAIC$_{N=94}$ = 66153). Furthermore, in this model $\kappa_{MPH-selective}$ relates negatively to Listening Span (R = 0.22, p=0.036; *Appendix 5—figure 1F*), now that this model accounts for the MPH-induced Pavlovian bias variance. This negative correlation cannot explain the positive relation between the relation between working memory span and the effects of MPH on motivational bias, and as such further corroborates our conclusion that MPH does not simply increase the strength of the instrumental learning bias as a function of listening span.

In the main article, we report the subject-level parameter estimates in model space (*Figures 3* and *5*). Here we additionally report the untransformed parameter estimates (*Appendix 5—table 1*: subject-level, *Appendix 5—table 2*: top-level) and the confidence of top-level parameters deviating from 0 for each model (*Appendix 5—table 3*). In *Appendix 5—figure 2,3* we display the one-step-ahead predictions for the both the winning and non-winning base and MPH models.

**Appendix 5—table 1.** Untransformed subject-level parameter means (SD).

| | Base models | | | | | Extended MPH models | | | | |
|---|---|---|---|---|---|---|---|---|---|---|
| | **M1** | **M2** | **M3a** | **M3b** | **M4** | **M5a** | **M5b** | **M5c** | **M6a** | **M6b** |
| $\rho$ | 3.4 (1.2) | 3.4 (1.2) | 3.3 (1.2) | 3.2 (1.1) | 3.2 (1.1) | 3.2 (1.1) | 3.2 (1.1) | 3.7 (1.3) | 3.2 (1.1) | 3.7 (1.3) |
| $\varepsilon$ | −3.7 (1.6) | −3.7 (1.5) | −3.6 (1.5) | −3.4 (1.5) | −3.5 (1.4) | −3.4 (1.4) | −3.5 (1.4) | −4.0 (1.7) | −3.4 (1.3) | −4.0 (1.7) |

*Appendix 5—table 1 continued on next page*

*Appendix 5—table 1 continued*

| | Base models | | | | | Extended MPH models | | | | |
|---|---|---|---|---|---|---|---|---|---|---|
| | **M1** | **M2** | **M3a** | **M3b** | **M4** | **M5a** | **M5b** | **M5c** | **M6a** | **M6b** |
| $b$ | | −0.2 (0.5) | −0.2 (0.5) | −0.04 (0.6) | −0.05 (0.5) | −0.06 (0.5) | −0.07 (0.5) | −0.13 (0.5) | −0.08 (0.5) | −0.15 (0.5) |
| $\pi$ | | | 0.5 (0.8) | | 0.15 (0.7) | 0.13 (1.0) | 0.18 (0.7) | 0.04 (0.7) | 0.17 (1.1) | 0.16 (0.9) |
| $\kappa$ | | | | 1.7 (1.3) | 1.2 (0.8) | 1.1 (0.7) | 1.1 (0.8) | 1.2 (0.8) | 1.1 (1.0) | 1.0 (0.7) |
| $\pi_{MPH}$ | | | | | | 0.08 (1.2) | | | 0.09 (1.5) | −0.19 (1.2) |
| $\kappa_{MPH\text{-}selective}$ | | | | | | | 0.09 (0.8) | | 0.22 (1.1) | |
| $\kappa_{MPH\text{-}diffuse}$ | | | | | | | | −5.9 (0.7) | | −5.7 (0.5) |

**Appendix 5—table 2.** Untransformed top-level parameter means (SD).

| | Base models | | | | | Extended MPH models | | | | |
|---|---|---|---|---|---|---|---|---|---|---|
| | **M1** | **M2** | **M3a** | **M3b** | **M4** | **M5a** | **M5b** | **M5c** | **M6a** | **M6b** |
| $\rho$ | 3.4 (0.2) | 3.4 (0.2) | 3.3 (0.2) | 3.2 (0.16) | 3.2 (0.15) | 3.2 (0.15) | 3.2 (0.15) | 3.7 (0.18) | 3.2 (0.15) | 3.7 (0.17) |
| $\varepsilon$ | −3.7 (0.2) | −3.7 (0.2) | −3.5 (0.2) | −3.4 (0.19) | −3.4 (0.19) | −3.4 (0.19) | −3.4 (0.19) | −4.0 (0.22) | −3.3 (0.18) | −3.9 (0.22) |
| $b$ | | −0.2 (0.1) | −0.2 (0.1) | −0.04 (0.07) | −0.05 (0.06) | −0.06 (0.06) | −0.07 (0.06) | −0.13 (0.06) | −0.08 (0.06) | −0.15 (0.06) |
| $\pi$ | | | 0.5 (0.1) | | 0.15 (0.09) | 0.13 (0.12) | 0.18 (0.09) | 0.04 (0.09) | 0.17 (0.13) | 0.16 (0.11) |
| $\kappa$ | | | | 1.65 (0.21) | 1.2 (0.15) | 1.1 (0.14) | 1.1 (0.17) | 1.16 (0.15) | 1.09 (0.18) | 1.00 (0.13) |
| $\pi_{MPH}$ | | | | | | 0.08 (0.14) | | | 0.09 (0.17) | −0.19 (0.14) |
| $\kappa_{MPH\text{-}selective}$ | | | | | | | 0.09 (0.19) | | 0.22 (0.25) | |
| $\kappa_{MPH\text{-}diffuse}$ | | | | | | | | −5.9 (0.24) | | −5.7 (0.21) |

**Appendix 5—table 3.** Confidence/probability that top-level parameter is larger than 0.

| | Base models | | | | | Extended MPH models | | | | |
|---|---|---|---|---|---|---|---|---|---|---|
| | **M1** | **M2** | **M3a** | **M3b** | **M4** | **M5a** | **M5b** | **M5c** | **M6a** | **M6b** |
| $\rho$ | 1.00 | 1.00 | 1.00 | 1.00 | 1.00 | 1.00 | 1.00 | 1.00 | 1.00 | 1.00 |
| $\varepsilon$ | 0.00 | 0.00 | 0.00 | 0.00 | 0.00 | 0.00 | 0.00 | 0.00 | 0.00 | 0.00 |
| $b$ | | 0.00 | 0.00 | 0.29 | 0.17 | 0.13 | 0.12 | 0.01 | 0.09 | 0.01 |
| $\pi$ | | | 1.00 | | 0.96 | 0.87 | 0.98 | 0.70 | 0.91 | 0.91 |
| $\kappa$ | | | | 1.00 | 1.00 | 1.00 | 1.00 | 1.00 | 1.00 | 1.00 |
| $\pi_{MPH}$ | | | | | | 0.72 | | | 0.71 | 0.09 |
| $\kappa_{MPH\text{-}selective}$ | | | | | | | 0.67 | | 0.81 | |
| $\kappa_{MPH\text{-}diffuse}$ | | | | | | | | 0.00 | | 0.00 |

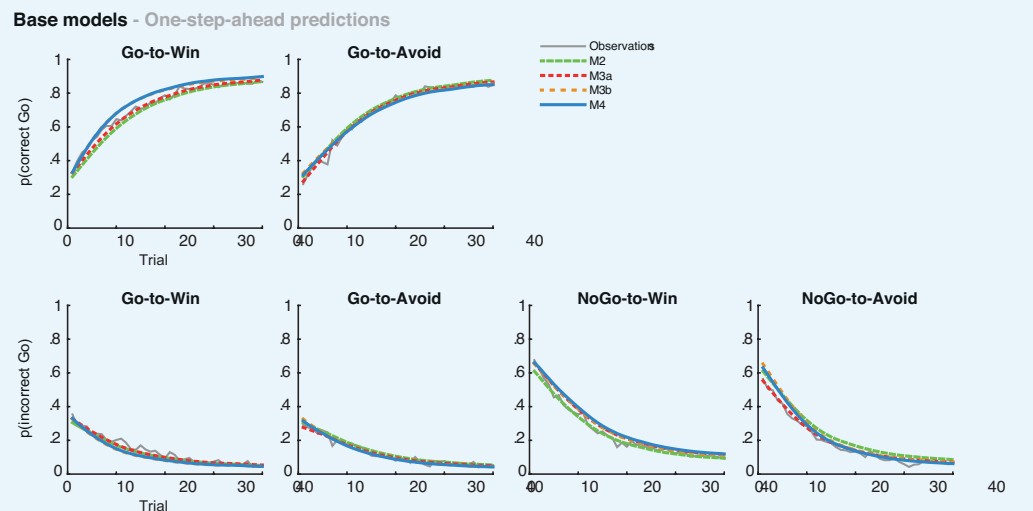

**Appendix 5—figure 2.** Average one-step-ahead predictions for the base models M2-4 overlaid on the observations in grey. The one-step-ahead predictions indicate the action probabilities as predicted by the model, using each subject's actual choices and outcomes.

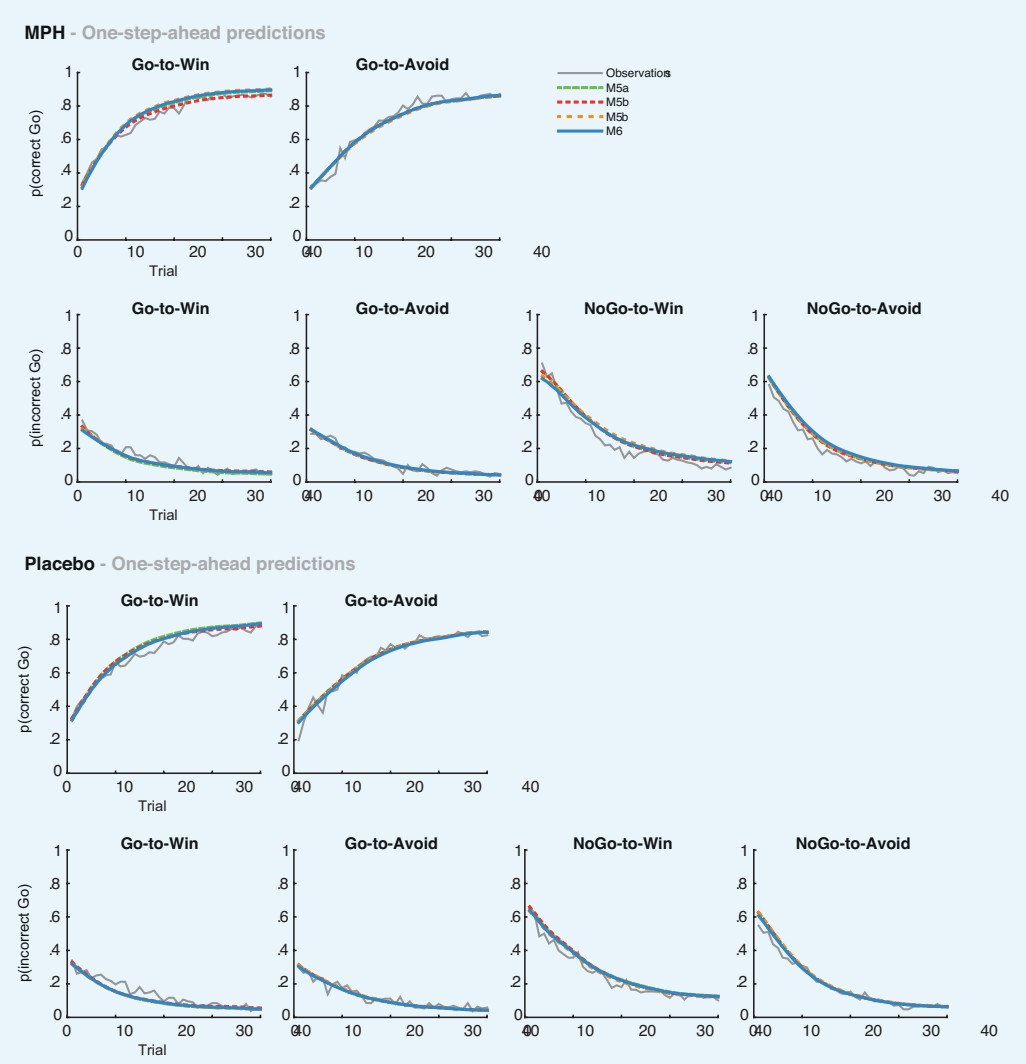

**Appendix 5—figure 3.** Average one-step-ahead predictions for the extended MPH models M5-6 overlaid on the observations in grey. The one-step-ahead predictions generate the action probability of each choice, based on the history of the subject's actual choices and outcomes preceding the choice. The predictions are separately plotted for MPH (top) and placebo (bottom). We observed no main effect of MPH on the motivational bias (i.e. more Go to Win cues relative to Avoid cues). Accordingly, all models make highly similar predictions under MPH and placebo across the group.

We refer to the *Decision Letter* and *Author Response* for a discussion on the potential confound of asymmetric reward/punishment sensitivities, where we show control analyses that speak against this potential confound.

