## [Decision Letter]

Thank you for submitting your article "Catecholaminergic challenge uncovers distinct Pavlovian and instrumental mechanisms of motivated (in)action" for consideration by *eLife*. Your article has been reviewed by three peer reviewers, one of whom, Naoshige Uchida (Reviewer #3), is a member of our Board of Reviewing Editors, and the evaluation has been overseen by Sabine Kastner as the Senior Editor. The following individual involved in review of your submission has agreed to reveal their identity: Sam Gershman (Reviewer #2).

The reviewers have discussed the reviews with one another and the Reviewing Editor has drafted this decision to help you prepare a revised submission.

Summary:

In this paper, the authors tested human subjects in a novel motivational Go/NoGo task (an extension of Guitart-Masip et al., 2012 task) in order to disentangle the impact of reward and punishment on instrumental learning from Pavlovian bias. Although the previous study had only one type of go response, the present study included two types of go responses (left versus right buttons), in order to test how reward and punishment affected specific instrumental behaviors. The authors tested a large number of subjects (N>100) in this task and also examined the effect of catecholamine uptake inhibitor (methylphenidate) on behavior.

All the reviewers found this study interesting and important. The multiple Go options and the pharmacological manipulation (MPH) are indeed potentially very powerful to further disentangle the Pavlovian system from the instrumental learning system. This is important because the previous study (Guitart-Masip et al., 2014) did not explicitly distinguish Pavlovian versus instrumental effects because the observed effect could be explained by reinforcement of a specific action (instrumental effect).

However, the reviewers pointed out various concerns that may affect the interpretation of the results. We would like to see your response to the following points:

Major points:

1) There appears to be a discrepancy between the best fitted model's simulation results (Figure 3) and the behavioral data (Figure 2). It is not clear whether the model's key predictions are supported by the data (although the authors performed extensive model comparisons).

1A) The key prediction of the biased instrumental learning model (the best fitted model; M4 with κ) is "faster learning for Go-to-Win" and "slower learning for Go-to-Avoid", as the former is driven by Go-Reward and the latter is driven by No-Go-Punishment. Is this right? This can be seen in the model simulation results (Figure 3), but not clear in the data (Figure 2). In fact, the overall difference in data (Figure 2) appears to be largely due to the initial difference (t=0, Figure 2), rather than the difference in learning rates. The learning for Go-to-Avoid looks a little faster than for Go-to-Win. Why?

1B) Shouldn't the initial difference between Win and Avoid in data (Difference between Green and Red at t=0 in Figure 2) be captured by the Pavlovian bias term (*π*) in the model? However, the initial difference in the simulation seems to be significantly smaller (Figure 3). In fact, the impact of the Pavlovian bias (*πV(s) =0.2*0.5=0.1* --- Equation 3) seems to be much smaller than the impact of rewards due to the large reward sensitivity.

2) One very strange thing about the best fitted parameters (Figure 3, Figure 5) is the very large sensitivity (*ρ*=40 or more), in contrast to what previous similar studies suggest (e.g. Huys et al., 2011, where they reported sensitivity = 3). This could suggest that the behavior is in fact extremely deterministic (one reward can swing the decision probability from 0 to 1). If this is the case, the stochastic feature of the model results (Figure 2) is not driven by the usual sense of learning but the stochasticity of the reward contingency (80% vs 20%), or even possibly by the sum of step functions (Gallistel et al., 2004). This very large sensitivity makes the model results very hard to interpret. It would be nice if the individual simulation traces are shown, in addition to the mean, so that we can see how the model actually behaves.

3) The authors perform their model selection by comparing different models that gradually incorporate additional parameters but they do so in one particular order. To establish the main conclusion, it is important to demonstrate the existence of both Pavlovian and instrumental effects. To do this, the authors should demonstrate the following. First, the authors should show that the model that includes all the parameters (model 4) is significantly better than the model that lacks only the Pavlovian or instrumental bias parameter (i.e. the model that lacks either b or κ). The current analysis only addresses this for κ. In other words, the authors should show that the model that does not include the b parameter performs worse than the full model (model 4) in order to prove the significance of b (Pavlovian effect).

4) There is a potential confound, due to a previously reported asymmetry between the reward and the punishment sensitivities (e.g. Huys et al., 2011). Since that the learning rate and the sensitivity are closely related with each other, and that the authors did not explore the asymmetric sensitivity, the assumed learning asymmetry (Equation.4) could be better altered by the previously reported asymmetric sensitivity that is independent of actions.

5) Reverse inference from working memory capacity and trait impulsivity to dopamine seems problematic, since these individual differences are presumably multi-factorial. One reviewer was surprised that the authors did not use something that is potentially less ambiguously related to baseline dopamine levels, such as spontaneous eyeblink rates. Relatedly, one of the authors (Frank) has suggested that working memory may instantiate a separate learning mechanism linked to prefrontal dopamine levels (Collins & Frank, 2012; Collins et al., 2014). This appears to complicate the interpretation of working memory capacity in this task, and at least requires some comment. We could also see the value of doing more modeling in this vein, using the models developed by Anne Collins, but since the modeling in this paper is quite extensive, we would be happy with leaving this as a task for future work.

[Editors' note: further revisions were requested prior to acceptance, as described below.]

Thank you for resubmitting your work entitled "Catecholaminergic challenge uncovers distinct Pavlovian and instrumental mechanisms of motivated (in)action" for further consideration at *eLife*. Your revised article has been favorably evaluated by Sabine Kastner (Senior editor), a Reviewing editor, and two reviewers.

The manuscript has been improved but there are some remaining issues that need to be addressed before acceptance, as outlined below:

All the reviewers thought that the authors have done a great job addressing the previous concerns and the manuscript is greatly improved. However, Reviewer #1 found that some clarifications are necessary. We therefore would like to see your response before proceeding for publication. Please respond to the two remaining points raised by Reviewer #1, as appended below:

Reviewer #1:

I appreciate the authors' effort to address our concerns. I think the paper became more informative and accessible. I am just a bit confused with a couple of points. It would be nice if the authors could clarify them in the final version.

I detail my confusions and then I write a suggestion.

1) In new Figure 3, the authors show that "The impact of the Pavlovian bias (*π*) on choice decreases over time." And then "We note that the initial difference in Go responding between Win and Avoid trials is somewhat underestimated (trial 1-2). This is likely the result from the decreasing impact of the Pavlovian bias over time (B).". This logic confuses me. If the impact of the Pavlovian bias is the largest in the first few trials, how should we expect the underestimation in those trials? I understand the argument of the likelihood reflecting the fitting performance of all trials; but it is not clear to me how the decreasing impact justifies the initial underestimation. Also it is not clear to me how Figure 13 supports that " This discrepancy (p(Go)Win-Avoid) is largely constrained to the first two trials and is absent in later trials." For example, p(Go)Win and p(Go)Avoid seem to become smaller over trials in the data, but in the model it doesn't seem to be. (I think the discrepancy on each trial is easy to quantify if authors wish to)

2) In response to our point 2, the authors claim that "the effective updating of the Q-value is the same order of magnitude as the Pavlovian influence on the action weights" However, I am not sure about this, because "The effective updating of the Q-value would be +.34 and +.55 for a rewarded NoGo and a rewarded Go response respectively" and the Pavlovian bias is 0.06 (-0.06) for Win (Avoid) trials (Equation 3). I don't think the relative effect is relevant here, because only one of them appears on each trial. Then I think it would be fair to say at least that the update +0.55 is an order of magnitude larger than the Pavlovian bias -0.06. (For the other point of the large sensitivity, I understand the authors response. The learning rate now became much smaller than the original version thanks to the median.)

For both points 1 and 2 above, I think what has been confusing me is that we don't see strong effects of the Pavlovian bias in the mean traces (or comparing the mean estimates). But what it really matters is the wide distribution of the Pavlovian bias (Figure 3—figure supplement 2). In fact, Figure 3—figure supplement 2 seems to clarify my confusions 1 and 2. So if the authors agree with my interpretation, I would suggest to stress in the paper that 1) the Pavlovian bias doesn't appear to be strong in the mean estimates (mean traces), but 2) the bias is so strong for a significant number of subjects (Figure 3—figure supplement 2) that it improved the overall fitting. It'd be nice if the authors could clarify the wide distribution of Pavlovian bias is consistent with previous studies.

Reviewer #2:

I think the authors have done a very thorough job addressing the issues raised in the reviews.

---

## [Author Response]

*Major points:*

*1) There appears to be a discrepancy between the best fitted model's simulation results (Figure 3) and the behavioral data (Figure 2). It is not clear whether the model's key predictions are supported by the data (although the authors performed extensive model comparisons).*

The reviewers comment on the discrepancy between the data and the model simulations, which was most striking for the Win cues; the model simulations overestimated the Go responding for both the Go-to-Win and NoGo-to-Win cues. This comment triggered us to look further into the model simulations. This led to three important observations/updates, which we feel are important to include in the paper, and add substantially to the understanding of the modelled dynamics. We will first discuss these changes before replying to the specific comments below.

Our first observation is a rather intricate point regarding the effect of transformation of parameters. We had originally implemented the instrumental learning bias such that its effect was symmetric in sampling space (prior to inverse logit transformation that enforces [0 1] bounds on the learning rate ε, with the intention to model a symmetric learning bias in the data. Under this implementation, the effect of the learning bias is only symmetric in model space if the modeled learning rate is 0.5:

Author response image 1.**DOI:**
http://dx.doi.org/10.7554/eLife.22169.028

Given that the learning rates in this study were <0.5, the increase in learning rate for rewarded Go was much larger than the decrease in learning rate for the punished NoGo. A priori we did not hypothesize an asymmetric effect of the instrumental learning bias, let alone that slow learners would necessarily be more affected by reward than punishment, and vice versa for quick learners. To correct for the unforeseen consequences of the original implementation, we revised the implementation of the instrumental learning bias (κ) such that the effect for reward and punishment is symmetrical in model space, i.e. after transformation. Specifically, we only applied the transformation in the punishment domain (to ensure the biased learning rate could not become negative) and applied the modeled learning bias directly to the reward domain:

Author response image 2.**DOI:**
http://dx.doi.org/10.7554/eLife.22169.029

Also see the applied formulas:

ε_0_= inv.logit(ε)ε_punished NoGo_= inv.logit(ε – κ)*previously:*ε_rewarded Go_= inv.logit(ε + κ)*revised:*ε_rewarded Go_= ε_0_ + (ε_0_- ε_punished NoGo_)

We then verified using model simulations that simulated data matched subjects' performance more closely under this new implementation. Most importantly, simulations now show less exaggerated learning of rewarded Go relative to the previous implementation (see Figure 12). The results of model comparison did not change: this updated M4 still outperformed M1-3. Because this model is a better reflection of our hypothesis (no a priori reason to believe that effects of κ are asymmetric), we replaced M4 with this updated version throughout the manuscript (figures, text), and added the following to the procedure section:

“To ensure that the effect of κ on ε (Eq.4) was symmetrical in model space (i.e. after sigmoid transformation to ensure [0 1] constraint), ε was computed as:

ε= {ε0=inv.logit(ε)εpunished NoGo=inv.logit(ε−κ)εrewarded Go=ε0+(ε0−εpunished NoGo) Eq. 5

Relatedly, we now realized that the inverse logistic transformation and exponential transformation (for the [0 1] and positivity constraint respectively) may result in a skewed distribution of the parameter estimates in model space, given that parameters were estimated under Gaussian priors in sampling space. For skewed, non-normal distributions, the median is a more suitable centrality measure than the mean. Therefore, we now report the parameter median as centrality measure and the 25-75 percentiles as dispersion measure throughout the manuscript (consistent with reporting by Guitart-Masip et al., 2014). See Table 1 and 2.

Note that we have re-labeled the extended MPH models (M5a, b, c), to refer to the models that include a single mechanism (i.e. parameter) to explain the effects of MPH.

Second, in our original implementation, we did 1000 simulations with each subject's parameter means. We now updated this to properly make use of the full posterior distribution of sampled parameters per subject rather than the parameter means. In other words, we performed one simulation per sampled parameter combination, resulting in 4000 simulations per subject. This respects the Bayesian nature of the sampling procedure, and takes into account the uncertainty of parameter estimates as a “posterior predictive check”.

Third, all learning effects depend on the experienced choices and outcomes. We therefore now also include so-called 'one-step-ahead' predictions, which more closely resemble the observed data compared to the model simulations (Figure 12). We included both absolute fit methods in the manuscript and explained our rationale in the Results section:

“We used two approaches to compute the post hoc absolute model fit, namely data simulation and "one-step-ahead" model predictions. […] We used both methods as the strongest test providing converging evidence that the models could capture the observed results.”

and in the procedure section:

“Having established the winning model, we used two absolute model fit approaches to confirm that the winning model captures the effects of interest; the post-hoc absolute-fit approach (also called one-step-ahead prediction) and posterior predictive model simulation approach (Steingroever and Wagenmakers, 2014). […]Averaging over repetitions also minimizes effects of randomness due to the stochastic nature of the choice simulation.”

Author response image 3.Behavioural observations and model predictions of winning base model M4.(**A**) Data averaged over placebo and MPH. Go responding is higher for Win than Avoid cues, independent of the action requirements. (**B**) Model M4 simulations from the original manuscript; the instrumental learning bias (κ) was symmetrical prior to [0 1] constraint, resulting in disproportionally strong Go-to-Win learning. (**C**) Revised model M4 simulations with effects of κ symmetrical in the reward and punishment domain better match the data. (**D**) One-step-ahead predictions for the revised model M4. The one-step-ahead predictions, which take into account all preceding choices and outcomes for predicting each, match the data even closer than the model simulations.**DOI:**
http://dx.doi.org/10.7554/eLife.22169.030

*1A) The key prediction of the biased instrumental learning model (the best fitted model; M4 with κ) is "faster learning for Go-to-Win" and "slower learning for Go-to-Avoid", as the former is driven by Go-Reward and the latter is driven by No-Go-Punishment. Is this right? This can be seen in the model simulation results (Figure 3), but not clear in the data (Figure 2). In fact, the overall difference in data (Figure 2) appears to be largely due to the initial difference (t=0, Figure 2), rather than the difference in learning rates. The learning for Go-to-Avoid looks a little faster than for Go-to-Win. Why?*

Indeed, these are the two key predictions of the biased instrumental learning model. We agree with the reviewers that there was a discrepancy between the simulations and the data, which triggered the updates discussed above. Now that we have ensured that κ has symmetrical effects on the learning rate (i.e. faster learning to go to win, and slower learning to go to avoid), and we have used one-step-ahead predictions to account for sequential effects, this difference has largely disappeared and the simulated data is much closer to the observed data (compare asymmetric and symmetric κ simulations in Figure 13).

Regarding the question whether these modelled differences in learning rate can be observed directly in the data, we agree that this a) looked a lot more exaggerated in the simulations than in the data and b) is not immediately obvious from the averaged data plot.

Regarding, the first point, the discrepancy between the model and data has disappeared under the revised implementation of κ: the effect of the difference in learning rates has become subtle for the model simulations and one-step-ahead predictions.

Regarding the second point, we agree with (the later point of) the reviewers that individual traces, in addition to averages, may be informative for the readers as well. We therefore now also include trial-by-trial choice probabilities (averaged over cues and sessions) for the different conditions (Figure 2—figure supplement 1). Illustrative of the faster learning in the Go-to-win conditions is that more individual subjects reach ceiling performance sooner. While this gives a qualitative illustration of these learning effects, we have added a further illustration of the differential learning rates in the one-step-ahead predictions: We separately plotted the predictions and data of subjects with relatively strong (top 33%) versus weak (bottom 33%) parameter estimates (Figure 3—figure supplement 2). As expected, the model shows steeper Go-to-Win than Go-to-Avoid curves especially for the subjects with relatively strong instrumental learning bias estimates. Also interestingly, when we correlate the ability of biased learning rate κ to explain the motivational bias in Go responding, we see that this increases over time, as expected for the cumulative effects of biased learning (see also comment 1B, and new Figure 3). We believe these figures provide valuable insight into the behavioural dynamics related to the parameters of interest and included the figures in the manuscript.

*1B) Shouldn't the initial difference between Win and Avoid in data (Difference between Green and Red at t=0 in Figure 2) be captured by the Pavlovian bias term (π) in the model? However, the initial difference in the simulation seems to be significantly smaller (Figure 3). In fact, the impact of the Pavlovian bias (πV(s) =0.2*0.5=0.1 --- Equation 3) seems to be much smaller than the impact of rewards due to the large reward sensitivity.*

We agree with the reviewers that the initial difference in Go responding between Win and Avoid trials is underestimated, particularly in M4. However, regarding your first point that the Pavlovian bias parameter should capture the average / global Pavlovian bias, i.e. the difference between the Win and Avoid conditions, we would like to note that this is a little more subtle: While the Pavlovian bias parameter is the only parameter that can capture the initialdifference, it is important to note that its estimate is based on performance on all trials. In other words, the initial difference in probability of Go responses, does not necessarily reflect the constant impact of cue valence on choice. In later trials, as subjects learn the instrumental values, the impact of the Pavlovian bias may go down, resulting in on average lower Pavlovian bias relative to trial 1 (reflecting, perhaps, the relative confidence in the Pavlovian and instrumental systems, cf. work by Daw et al. on uncertainty-based competition, Nature Neuroscience, 2005, a factor not captured by our models). Indeed, when we compare the model estimates to the data from trial 3 onwards, there is much smaller difference between data and simulations (Figure 13). In other words, the discrepancy between simulated and real data is only present on very early trials, when subjects have no prior experience to go on and are very likely purely driven by their Pavlovian responses. In contrast, model M3a attributes the variance that results from biased instrumental learning to the Pavlovian bias parameter, inflating the Pavlovian bias estimates. As a result of these inflated estimates, this model will be better able to capture the initial difference in Go responding to Win vs. Avoid cues (see also one-step-ahead predictions in Figure 3—figure supplement 2 at comment #11).

Author response image 4.Observed behavior and one-step-ahead predictions excluding trials 1-2.The model predictions show more pronounced p(Go)_Win-Avoid_ on the first trials than the observations (Figure 12). This discrepancy is largely constrained to the first two trials and is absent in later trials, as can be seen more easily when omitting the first 2 trials for illustration.**DOI:**
http://dx.doi.org/10.7554/eLife.22169.031

Related to this point about the potential temporal dynamics of the impact of the Pavlovian bias, previous studies have used cues where subjects had to learn the Pavlovian value (cf. Guitart-Masip ea. 2012), and fitted models in which Pavlovian values were learnt and allowed to change over time. In the current study, we had opted for modelling a fixed Pavlovian bias, because valence was signalled by the coloured edge (red/green) of the cues. Nonetheless, triggered by this reviewers' comment, we fitted a control model in which Pavlovian values are learnt. Model comparison clearly favoured model M3a with fixed Pavlovian bias over this model with learned Pavlovian values (ΔWAIC = -318), in line with the instructed cue valence in our task. Thus, model selection suggests more evidence for a fixed Pavlovian bias than Pavlovian value learning. Note that a model with Pavlovian learning predicts an increasing reward bias when actions are learnt (i.e. rewards are obtained), but a decreasing punishment bias as punishments are avoided. Thus, this model comparison does not test this possibility that the impact of the Pavlovian bias decreases over time, which might result in a relative under-estimation of the initial Pavlovian bias. Triggered by this hypothesis, we assessed the correlation over subjects between the parameter estimate and the predicted motivational bias (i.e. probability to make a go response to 'win' relative to 'avoid' cues), over time. This correlation shows the relative contribution of that parameter in explaining the behavioural bias in the model. We included this correlation in Figure 3 and discuss the reason for the initial underestimation.

Finally, regarding your second point about the relative effect size of the Pavlovian bias and RL prediction-error based update, these are the same order of magnitude, as we discuss in more detail in our reply to the comment 2 below.

*2) One very strange thing about the best fitted parameters (Figure 3, Figure 5) is the very large sensitivity (ρ=40 or more), in contrast to what previous similar studies suggest (e.g. Huys et al., 2011, where they reported sensitivity = 3). This could suggest that the behavior is in fact extremely deterministic (one reward can swing the decision probability from 0 to 1). If this is the case, the stochastic feature of the model results (Figure 2) is not driven by the usual sense of learning but the stochasticity of the reward contingency (80% vs 20%), or even possibly by the sum of step functions (Gallistel et al., 2004). This very large sensitivity makes the model results very hard to interpret. It would be nice if the individual simulation traces are shown, in addition to the mean, so that we can see how the model actually behaves.*

The feedback sensitivity (*ρ*) is indeed quite large (median=32.5). However, its impact on behavior is restricted by the low learning rates (ε), as these are multiplied, c.f. Eq.2:

Qt(at,st)= Qt−1(at,st)+ \varepsilon (ρrt−\ Qt−1(at,st))Eq.2

Consider for example the update following a reward on the first trial, using the medians of the winning base model M4 (ρ=32.5, ε_0_=.021, ε_rewarded Go_=.034). The effective updating of the Q-value would be +.34 and +.55 for a rewarded NoGo and a rewarded Go response respectively:

16.25 +.021 * (32.5*1 – 16.25) = 16.6 *if a = NoGo* Eq.2

16.25 +.034 * (32.5*1 – 16.25) = 16.8 *if a = Go*

These increases in Q-values would increase the action probability from.33 to.42 (*a*=NoGo) and.46 (*a*=Go) according to Eq. 1 (we disregarded the Pavlovian bias here for simplicity). Thus, the high feedback sensitivity does not result in extremely deterministic behavior, due to the low learning rate. Moreover, feedback sensitivity is strongly anti-correlated with the learning rate (R=-.81, *p* <.001) across subjects, such that the impact of higher feedback sensitivity estimates is restricted by lower learning rates. Note that this dependency between these two parameters is well known, and is not problematic here because we are not interested in estimating these independently. We now included the correlations between parameters in Figure 3—figure supplement 3 and Figure 5—figure supplement 2.

In relation to comment #1b, note that the effective updating of the Q-value is the same order of magnitude as the Pavlovian influence on the action weights:

πV(s)Eq.30.12 * +0.5 = +.06*if s = Win cue*0.12 * -0.5 = -.06*if s = Avoid cue*Relative effect (win-avoid) =.12

To illustrate this point, as the reviewers suggest, we now include the individual data and simulation (one-step-ahead prediction) traces in the manuscript in Figure 3—figure supplement 1. These traces confirm that choice probabilities indeed change gradually in individual subjects. In fact, learning looks relatively slower compared to previous reports (e.g. Guitart-Masip et al., 2012), where most responses were learned within the first 10 trials. This is likely to have resulted from increased task difficulty caused by the additional response option (3 instead of 2) and double number of cues.

*3) The authors perform their model selection by comparing different models that gradually incorporate additional parameters but they do so in one particular order. To establish the main conclusion, it is important to demonstrate the existence of both Pavlovian and instrumental effects. To do this, the authors should demonstrate the following. First, the authors should show that the model that includes all the parameters (model 4) is significantly better than the model that lacks only the Pavlovian or instrumental bias parameter (i.e. the model that lacks either b or κ). The current analysis only addresses this for κ. In other words, the authors should show that the model that does not include the b parameter performs worse than the full model (model 4) in order to prove the significance of b (Pavlovian effect).*

We had originally not included the model comparison the reviewers request here, because the biased behaviour on the first trials (i.e. without any learning yet), could not have resulted from a learning bias alone. However, we agree that to strengthen our conclusions, particularly in light of the points we make in reply to comment 1B, we now added this model to our model space. Model fit is indeed better for the full base model (M4) compared with the proposed model without the Pavlovian bias (M3b, ΔWAIC=-615), providing evidence for the presence of both the Pavlovian response bias and biased instrumental learning. We now include this model in our manuscript (Figure 3; Table 1; see reply to comment 1).

*4) There is a potential confound, due to a previously reported asymmetry between the reward and the punishment sensitivities (e.g. Huys et al., 2011). Since that the learning rate and the sensitivity are closely related with each other, and that the authors did not explore the asymmetric sensitivity, the assumed learning asymmetry (Equation 4) could be better altered by the previously reported asymmetric sensitivity that is independent of actions.*

There is indeed previous evidence for asymmetries in sensitivity to rewards and punishments. We explored using both model simulations and model fitting whether such an asymmetry could potentially confound our conclusions, and we conclude that it cannot. We would also like to note that separate reward/punishment sensitivities were not supported by model comparison in a previous comparable task either (Guitart-Masip et al., 2012).

1) Model simulation. Asymmetric reward/punishment sensitivity is insufficient in explaining the observed motivational bias in action, i.e. the coupling of reward to action and punishment to inaction. We illustrate this point using simulations based on our basic model without any motivational bias parameters (M2), where we varied the reward and punishment sensitivity independently (Figure 14). Higher reward sensitivity improves both Go and NoGo responding for Win relative to Avoid cues, and vice versa for higher punishment sensitivity. In contrast, in our data we observe that subjects are better at learning to Go and worse at learning to NoGo for Win cues relative to Avoid cues. As the plots in Figure 14 illustrate, feedback sensitivity is not able to explain this motivational asymmetry in action, even when reward and punishment sensitivity can vary independently.

Author response image 5.Model predictions of asymmetrical reward/punishment sensitivity (*ρ*_rew/pun_).Here we simulate data using model M2, which consists of a simple reinforcement learning model and a go bias parameter. We used the fitted group-level parameters (learning rate=.024; feedback sensitivity= 30; go bias = -.2) and averaged over 1000 simulations. A. Model predictions of model M2 without asymmetric sensitivity. The model predicts no differences between Win and Avoid cues, i.e. Go responding matches for Win and Avoid cues. B. Model predictions of model M2 + higher reward sensitivity (reward sensitivity = 40; punishment sensitivity= 20). Stronger reward sensitivity predicts incorrectly that subjects make more NoGo responses for the NoGo-to-Win than NoGo-to-Avoid cues. C. Model predictions of model M2 + higher punishment sensitivity (reward sensitivity = 20; punishment sensitivity= 40). Stronger punishment sensitivity predicts incorrectly that subjects make more Go responses Go-to-Avoid than Go-to-Win cues.**DOI:**
http://dx.doi.org/10.7554/eLife.22169.032

2) Model comparison. To address the reviewers’ concerns further, we explored whether separate reward/punishment sensitivities could explain the data better than the instrumental learning bias in our winning base model M4. To this end, we fitted a control model with independent reward/punishment sensitivity instead of the instrumental learning bias (RL_rew/pun_ + go bias + Pavlovian response bias). The reward and punishment sensitivities did not significantly differ across the group (Δ*ρ*_rew/pun_: 30.3% of posterior distribution > 0). Crucially, model evidence reduced for this control model relative to model M4 (RL + go bias + Pavlovian response bias + instrumental learning bias; ΔWAIC=+60). The reduced model evidence indicates that the data are better explained by the instrumental learning bias than by separate reward/punishment sensitivities, which is in line with the simulations in Figure 14 that the motivational bias cannot be understood in terms of asymmetric reward/punishment sensitivities.

Altogether, this speaks against a potential confound of asymmetric feedback sensitivity. In Appendix 5, we refer to this section for interested readers for whom the same question may arise:

We refer to the Decision Letter and Author Response for a discussion on the potential confound of asymmetric reward/punishment sensitivities, where we show control analyses that speak against this potential confound.

*5) Reverse inference from working memory capacity and trait impulsivity to dopamine seems problematic, since these individual differences are presumably multi-factorial. One reviewer was surprised that the authors did not use something that is potentially less ambiguously related to baseline dopamine levels, such as spontaneous eyeblink rates.*

We agree with the reviewers that working memory capacity is determined by many factors and does not only depend on striatal dopamine. When designing this study, we decided to use working memory capacity and trait impulsivity as a proxy for (striatal) dopamine, given that previous studies demonstrated a direct link (Cools et al., 2008; Landau et al., 2009; Buckholtz et al., 2010; Kim et al., 2014; Lee et al., 2009; Reeves et al., 2012) and a larger literature showed that working memory capacity predicts dopaminergic drug effects on cognition in a manner that is consistent with WM reflecting striatal dopamine (Cools et al., 2009, 2007; Frank and O’Reilly, 2006; Gibbs and D’Esposito, 2005; Kimberg et al., 1997; van der Schaaf et al., 2013; for a review see Frank & Fossela, 2011). Finally, in all honesty, we had not considered spontaneous eye blink rate when setting up this study, and this indeed would have been an interesting additional proxy, which we will certainly consider using in future studies.

We now revised the discussion of these points in the manuscript (note that regarding the point about including blink rate and other cautions for interpreting WM capacity results, we included these in our reply / revision for the next section):

Introduction section: “Finally, MPH prolongs the effects of catecholamine release by blocking the reuptake of catecholamines, without stimulating release or acting as a receptor (ant)agonist (e.g. Volkow et al., 2002). […]To assess these potential sources of individual variability in MPH effects, we took into account two measures that have been demonstrated with PET to relate to dopamine baseline function: working memory capacity for its relation to striatal dopamine synthesis capacity (Cools et al., 2008; Landau et al., 2009) and trait impulsivity for its relation to dopamine (auto)receptor availability (Buckholtz et al., 2010; Kim et al., 2014; Lee et al., 2009; Reeves et al., 2012), and collected a large sample (N=106) to expose individual differences.”

Results section: “Next, we asked whether acute administration of MPH altered the motivational bias. […] Importantly, both working memory span and trait impulsivity predict dopaminergic drugs effects on various cognitive functions (Clatworthy et al., 2009; Cools et al., 2009, 2007; Frank and O’Reilly, 2006; Gibbs and D’Esposito, 2005; Kimberg et al., 1997; van der Schaaf et al., 2013).”

Discussion section: “Individuals vary strongly in the extent to which MPH increases extracellular dopamine […] Together this may explain why the observed MPH effects covary with working memory span but not trait impulsivity.”

Relatedly, one of the authors (Frank) has suggested that working memory may instantiate a separate learning mechanism linked to prefrontal dopamine levels (Collins & Frank, 2012; Collins et al., 2014). This appears to complicate the interpretation of working memory capacity in this task, and at least requires some comment. We could also see the value of doing more modeling in this vein, using the models developed by Anne Collins, but since the modeling in this paper is quite extensive, we would be happy with leaving this as a task for future work.

Working memory processing involves both striatal and prefrontal dopamine, where striatal dopamine might relate particularly to updating and prefrontal dopamine to maintenance (e.g. Cools & D’Esposito, 2011). The Collins & Frank work assumes prefrontal DA contribution to maintenance which then complements the striatal RL process but it does not reject Frank’s earlier work implicating striatal DA in learning to update WM, which is aligned with the current interpretation (e.g., Frank & O’Reilly, 2006; Frank & Fossella 2011). The listening span task used here does not dissociate between updating and maintenance, though, as mentioned above, baseline span predicts dopaminergic drug effects not only on cognition but also on probabilistic RL function, consistent with the hypothesis that drug effects reflect modulation of striatal dopamine function.

A second possibility raised by the finding that drug effects depend on span is that they reflect modulation of working memory itself. We do not believe that this is a likely explanation of the relation between working memory capacity and the effects of MPH for two reasons:

First, there was no significant effect of baseline working memory on motivational bias under placebo conditions, but instead WM only predicted the impact of MPH. This strongly suggests that WM is predictive of the effects of MPH, rather than simply directly affecting performance (as we then would also have expected effects under placebo).

Second, we performed some further model fitting using the mixed RL / working memory models that the reviewers suggest to explore whether such working memory processes help to explain behaviour in this task. We extended model M2 (simple RL + go bias) with a forgetting component (cf. Collins & Frank, 2012; Collins et al., 2014), such that action values decay towards their initial values on every trial, as follows:

Q(a,s) = Q(a,s) + ф(Q _0_ – Q(a,s)).

Collins’ work has shown that forgetting is a property particularly within the WM system more than RL. Surprisingly, the forgetting component reduced model fit compared to M2 (ΔWAIC=+22), indicating trial-by-trial forgetting does not help to explain task performance, suggesting either that WM does not contribute substantially to this task, or that the load imposed on subjects was not high enough to reveal forgetting (Collins’ work specifically manipulates the set size to push subjects over their capacity limit), but in either case, there is no evidence that variations in WM are driving the effects we see directly. Given that this model does not help us to further explain performance, and as the reviewers also note, since the modeling in this paper is already quite extensive, we would prefer to not include these model comparisons in the paper.

We revised the discussion following the points above:

“The finding that drug effects depend on working memory is highly consistent with the hypothesis that they reflect modulation of striatal dopamine (c.f. Frank & Fossela, 2011). […] This is one of the major aims of our ongoing work.”

[Editors' note: further revisions were requested prior to acceptance, as described below.]

*Reviewer #1:*

1) In new Figure 3, the authors show that "The impact of the Pavlovian bias (π) on choice decreases over time." And then "We note that the initial difference in Go responding between Win and Avoid trials is somewhat underestimated (trial 1-2). This is likely the result from the decreasing impact of the Pavlovian bias over time (B).". This logic confuses me. If the impact of the Pavlovian bias is the largest in the first few trials, how should we expect the underestimation in those trials? I understand the argument of the likelihood reflecting the fitting performance of all trials; but it is not clear to me how the decreasing impact justifies the initial underestimation.

We apologize for the confusion that we caused here – we realized that we were making 2 points at the same time. First, the model underestimates the Pavlovian bias (i.e. p(Go)_Win –_ p(Go)_Avoid_) on the initial trials, while it overestimates the Pavlovian bias on later trials (as the reviewer points out below also; see Figure 3). Such an initial under-estimation and later over-estimation is the consequence of a constant parameter estimate but dynamically changing actual Pavlovian bias. We hypothesize (but cannot directly test this hypothesis in the current experimental setup) that the relative impact of Pavlovian and instrumental values is related to the relative confidence in each, in line with ideas from e.g. Daw et al. (2005, Nature Neuroscience) of uncertainty-based competition. We aim to test this hypothesis in future studies were we dynamically change the action-outcome contingencies to modulate confidence in the instrumental action values.

The second point is that notwithstanding the constant bias parameter, we do capture some of these dynamics as Figure 3 shows that the impact of the Pavlovian bias (π) on choice decreases over time. To make these points clear, we’ve made the following change (in bold) in the Figure 3 legend:

“[…] We note that the model somewhat underestimates the initial Pavlovian bias (i.e. difference in Go responding between Win and Avoid trials is, particularly trial 1-2), while it overestimates the Pavlovian bias on later trials. This is likely the result from the fact that while the modelled Pavlovian bias parameter (π) is constant over time, the impact of the Pavlovian stimulus values weakens over time, as the subjects’ confidence in the instrumental action values increases. Interestingly, notwithstanding the constancy of the Pavlovian bias parameter, we do capture some of these dynamics as Figure 3 shows that the impact of the Pavlovian bias on choice decreases over time.”

*Also it is not clear to me how Figure 13 supports that " This discrepancy (p(Go)Win-Avoid) is largely constrained to the first two trials and is absent in later trials." For example, p(Go)Win and p(Go)Avoid seem to become smaller over trials in the data, but in the model it doesn't seem to be. (I think the discrepancy on each trial is easy to quantify if authors wish to)*

In Figure 13, we aimed to illustrate that the directional discrepancy between the model and data (i.e. p(Go)_Win-Avoid | model_ < p(Go)_Win-Avoid |data_) is predominantly present on the first few trials. However, as the reviewer correctly points out, p(Go)_Win–Avoid_ decreases over trials in the data relative to in the model, such that the discrepancy has the opposite direction in later trials. We did not intent to argue that there is no absolute discrepancy in later trials, but rather that the directional discrepancy (p(Go)_Win-Avoid | model_ < p(Go)_Win-Avoid |data_) is predominantly present on the early trials, as we hope to now have also clarified in the previous point.

*2) In response to our point 2, the authors claim that "the effective updating of the Q-value is the same order of magnitude as the Pavlovian influence on the action weights" However, I am not sure about this, because "The effective updating of the Q-value would be +.34 and +.55 for a rewarded NoGo and a rewarded Go response respectively" and the Pavlovian bias is 0.06 (-0.06) for Win (Avoid) trials (Equation 3). I don't think the relative effect is relevant here, because only one of them appears on each trial. Then I think it would be fair to say at least that the update +0.55 is an order of magnitude larger than the Pavlovian bias -0.06. (For the other point of the large sensitivity, I understand the authors’ response. The learning rate now became much smaller than the original version thanks to the median.)*

We agree that it is indeed fair to say that the impact of the Pavlovian bias is up to an order of magnitude smaller than the effective updating. In the paper we did not discuss these proportional effects explicitly, but rather focused on the combined impact of learning rate and sensitivity, such that the low learning rate tones down the impact of the high sensitivity, as the reviewer also appreciates here. We now comment on the small mean effect of the Pavlovian bias in the manuscript, in line with the reviewer’s suggestions (see below).

*For both points 1 and 2 above, I think what has been confusing me is that we don't see strong effects of the Pavlovian bias in the mean traces (or comparing the mean estimates). But what it really matters is the wide distribution of the Pavlovian bias (Figure 3—figure supplement 2). In fact, Figure 3—figure supplement 2 seems to clarify my confusions 1 and 2. So if the authors agree with my interpretation, I would suggest to stress in the paper that 1) the Pavlovian bias doesn't appear to be strong in the mean estimates (mean traces), but 2) the bias is so strong for a significant number of subjects (Figure 3—figure supplement 2) that it improved the overall fitting. It'd be nice if the authors could clarify the wide distribution of Pavlovian bias is consistent with previous studies.*

We appreciate the reviewer’s suggesting, which we’ve incorporated in the relevant section:

“The Pavlovian bias parameter estimates (π) of the winning model M4 were positive across the group (96.4% of posterior distribution > 0). […] This strong inter-individual variability is consistent with previous reports, e.g. Cavanagh et al., (2013), who show that differences in the strength of the Pavlovian bias is inversely predicted by EEG mid-frontal theta activity during incongruent relative to congruent cues, putatively reflecting the ability to suppress this bias on incongruent trials.”